# Galvanic current activates the NLRP3 inflammasome to promote Type I collagen production in tendon

**Alejandro Peñin-Franch**[1], **José Antonio García-Vidal**[1,2], **Carlos Manuel Martínez**[1], **Pilar Escolar-Reina**[1,2], **Rosa M Martínez-Ojeda**[3], **Ana I Gómez**[1], **Juan M Bueno**[3], **Francisco Minaya-Muñoz**[4,5], **Fermín Valera-Garrido**[4,5,6], **Francesc Medina-Mirapeix**[1,2], **Pablo Pelegrín**[1,7]*

[1]Instituto Murciano de Investigación Biosanitaria IMIB-Arrixaca, Hospital Clínico Universitario Virgen de la Arrixaca, Murcia, Spain; [2]Department of Physical Therapy, University of Murcia, Murcia, Spain; [3]Laboratorio de Óptica, Instituto Universitario de Investigación en Óptica y Nanofísica, Universidad de Murcia, Murcia, Spain; [4]CEU San Pablo University, Madrid, Spain; [5]MVClinic Institute, Madrid, Spain; [6]Invasive Physiotherapy Department, Getafe C.F, Madrid, Spain; [7]Department of Biochemistry and Molecular Biology B and Immunology, University of Murcia, Murcia, Spain

**Abstract** The NLRP3 inflammasome coordinates inflammation in response to different pathogen- and damage-associated molecular patterns, being implicated in different infectious, chronic inflammatory, metabolic and degenerative diseases. In chronic tendinopathic lesions, different non-resolving mechanisms produce a degenerative condition that impairs tissue healing and which therefore complicates their clinical management. Percutaneous needle electrolysis consists of the application of a galvanic current and is an emerging treatment for tendinopathies. In the present study, we found that galvanic current activates the NLRP3 inflammasome and induces an inflammatory response that promotes a collagen-mediated regeneration of the tendon in mice. This study establishes the molecular mechanism of percutaneous electrolysis that can be used to treat chronic lesions and describes the beneficial effects of an induced inflammasome-related response.

**\*For correspondence:**
pablo.pelegrin@imib.es

## Editor's evaluation

This study investigated the mechanisms that underlie chronic tendinopathies. The authors identified the role of galvanic current on cellular death and inflammation in vitro and its association with tendinopathy in vivo. They showed that galvanic current induces NLRP3 inflammasome-driven low-grade inflammatory response, which promotes collagen-mediated regeneration of the tendon. This work may lead to a therapeutic strategy to treat chronic tendinopathies.

## Introduction

Galvanic current applied using a percutaneous needle is an emerging and minimally invasive technique that seeks to regenerate damaged tissues (*Valera-Garrido et al., 2014*). The needle used to apply galvanic current is guided, using ultrasound equipment, into affected soft tissues by direct visualization and therefore is able to stimulate locally the cells in the damaged area. Percutaneous needle electrolysis consists of applying a galvanic current through an acupuncture needle and thus combining mechanical and electrical stimulation of the tissue that results in a local controlled microtrauma. This microtrauma in turn generates a local inflammatory response that makes possible and

fosters the repair of the affected tissue (*Valera-Garrido and Minaya-Muñoz, 2019*). Galvanic current has been successfully used to repair chronic non-resolving lesions, such as tendinopathies developed after prolonged extreme exercise, which often establish a degenerative condition of the tissue that impairs healing and complicates clinical management (*Cook and Purdam, 2009*; *Regan et al., 1992*; *Soslowsky et al., 2000*). In randomized trials, anti-inflammatory therapies have shown to be ineffectual at treating these types of lesion (*Bisset et al., 2006*; *Coombes et al., 2013*) and application of galvanic currents by percutaneous needle alone has been found sufficient when it comes to regenerating the tissue (*Bubnov et al., 2013*; *Chellini et al., 2019*; *De-la-Cruz-Torres et al., 2020*; *Margalef et al., 2020*; *Valera-Garrido et al., 2020*; *Valera-Garrido et al., 2014*; *Valera-Garrido et al., 2013*).

Resolving inflammation promotes a response that recovers homeostasis by repairing tissues (*Medzhitov, 2008*). However, chronic non-resolving inflammation can induce continuous tissue regeneration and excessive accumulation of extracellular matrix components, which in turn leads to fibrosis of soft tissues (*Alegre et al., 2017*; *Borthwick et al., 2013*; *Gaul et al., 2021*; *Wynn, 2008*). Therefore, a balanced inflammatory response is required to recover homeostasis and successfully heal tissue (*Borthwick et al., 2013*; *Eming et al., 2017*; *Liston and Masters, 2017*; *Medzhitov, 2008*). In response to tissue damage, the nucleotide-binding oligomerization domain with leucine-rich repeat and pyrin domain containing 3 (NLRP3) inflammasome is specifically activated over other inflammatory pathways and coordinates an inflammatory response (*Broz and Dixit, 2016*; *Schroder and Tschopp, 2010*). NLRP3 inflammasome is a multiprotein complex induced in myeloid cells after the detection of damage- or pathogen-associated molecular patterns, including elevated concentrations of extracellular ATP, changes in extracellular osmolarity or detection of insoluble particles and crystals, such as uric acid crystals or amyloid deposition (*Amores-Iniesta et al., 2017*; *Compan et al., 2012*; *Heneka et al., 2013*; *Martinon et al., 2006*). These triggers specifically induce NLRP3 activation over other inflammasomes as they decrease intracellular $K^+$ concentration resulting in conformational change of NLRP3 structure and, therefore, active NLRP3 oligomers (*Munoz-Planillo et al., 2013*; *Tapia-Abellán et al., 2021*). Active NLRP3 oligomers recruit the accessory apoptosis-speck-like protein with a caspase recruitment and activation domain (ASC) that favors the activation of the inflammatory caspase-1 (*Boucher et al., 2018*; *Li et al., 2018*; *Schmidt et al., 2016*). Caspase-1 proteolytically processes immature pro-inflammatory cytokines of the interleukin (IL)–1 family to produce the bioactive form of IL-1β and IL-18 (*Broz and Dixit, 2016*; *Schroder and Tschopp, 2010*). Caspase-1 also processes gasdermin D protein (GSDMD), and its amino-terminal fragment (GSDMD$^{NT}$) oligomerizes in the plasma membrane forming pores, thus releasing IL-1β and IL-18 cytokines, as well as other intracellular content, including inflammasome oligomers (*Baroja-Mazo et al., 2014*; *Broz et al., 2020*).

Prolonged NLRP3 activation occurs in different chronic inflammatory, metabolic and degenerative diseases such as gout, type 2 diabetes or Alzheimer (*Daniels et al., 2016*; *Heneka et al., 2013*; *Masters et al., 2010*; *Martinon et al., 2006*), therefore selective small molecules that block NLRP3 are emerging as novel anti-inflammatory therapies (*Cocco et al., 2017*; *Coll et al., 2015*; *Tapia-Abellán et al., 2019*). However, in some pathological circumstances, a boost, rather than an inhibition, of NLRP3 would be more beneficial for reducing clinical complications, such as in immunosuppressed septic patients, who have high mortality rates due to secondary infections associated with a profound deactivation of the NLRP3 inflammasome (*Martinez-García et al., 2019*).

The detailed molecular mechanism behind percutaneous needle electrolysis inducing an inflammatory response has not been yet described, so we studied if NLRP3 inflammasome could be activated via galvanic currents and if doing so would lead to tissue regeneration. We found that applying galvanic current via percutaneous needle electrolysis activated the NLRP3 inflammasome and induced the release of IL-1β and IL-18 from macrophages. Mice deficient in NLRP3 failed to increase IL-1β in tendons after percutaneous needle electrolysis and ended up with reduced TGF-β and type I collagen deposition with high dispersion of collagen fibers, which in turn resulted in tendons with decreased resistance. All this indicates that the NLRP3 inflammasome plays an important role in the regenerative response of the tendon after application of therapeutically percutaneous needle electrolysis.

## Results

### Galvanic current enhances macrophage pro-inflammatory M1 phenotype

We initially designed and produced a device to apply galvanic current to adherent cultured cells in six-well cell-culture plates (*Figure 1—figure supplement 1*), this device allowed us to explore the effect of galvanic currents in mouse macrophages derived from bone marrow. Applying 12 mA of galvanic current twice for 6 s each time to LPS stimulated macrophages increase the expression of *Cox2* and *Il6* genes (*Figure 1A*). However, it did not affect LPS-induced *Il1b* or *Tnfa* pro-inflammatory gene expression (*Figure 1A*). Interestingly, meanwhile *Tnfa* expression was upregulated with galvanic current alone (*Figure 1A*), galvanic currents did not induce the expression of *Cox2*, *Il6* or *Il1b* genes on non-LPS-treated macrophages, or over IL-4-treated macrophages (*Figure 1A*). When macrophages were polarized to M2 by IL-4, galvanic currents decreased the expression of the M2 markers *Arg1*, *Fizz1*, and *Mrc1* (*Figure 1B*); however, this decrease was small and non-significant for the M2 marker *Ym1* (*Figure 1B*). These data suggest that galvanic current could enhance the pro-inflammatory signature of M1 macrophages whilst decreasing M2 polarization. We next studied the concentration of released pro-inflammatory cytokines from macrophages, and found that galvanic current did not increase the concentration of IL-6 or TNF-α released after LPS stimulation (*Figure 1C*), but that it did significantly augment the release of IL-1β in an intensity-dependent manner (*Figure 1C*). These data indicate that the increase in *Il6* and *Tnfa* gene expression detected at mRNA level does not transcribe to higher release of IL-6 and TNF-α over LPS treatment, but that galvanic current could potentially activate an inflammasome that induces the release of IL-1β.

### Galvanic current activates the NLRP3 inflammasome

Given that IL-1β release is increased by the activation of caspase-1 after the canonical or non-canonical formation of inflammasome (*Broz and Dixit, 2016*), we next studied the release of IL-1β induced by galvanic current in macrophages deficient in caspase-1 and –11 to avoid both the canonical and non-canonical inflammasome signaling. We found that *Casp1/11$^{-/-}$* macrophages fail to release IL-1β induced by galvanic current (*Figure 2A*). We then found that applying galvanic current to *Pycard$^{-/-}$* macrophages also failed to induce the release of IL-1β, which indicated that the inflammasome adaptor protein ASC is also needed to activate the inflammasome (*Figure 2A*). Since applying current could be considered a sterile danger signal, we next assessed the role of NLRP3 as an inflammasome sensor important for eliciting an immune response in sterile dangerous situations (*Broz and Dixit, 2016*; *Liston and Masters, 2017*). *Nlrp3$^{-/-}$* and the use of the specific NLRP3 inhibitor MCC950 (*Coll et al., 2015*; *Tapia-Abellán et al., 2019*) impaired the release of IL-1β induced by galvanic current (*Figure 2A and B*), demonstrating that the NLRP3 inflammasome is activated when galvanic current is applied. Similarly, galvanic current was also able to induce the release of IL-18 (*Figure 2C*), another cytokine dependent on the activation of the inflammasome. The use of MCC950 or macrophages deficient in NLRP3 failed to release IL-18 after galvanic current was applied (*Figure 2C*), which confirms that galvanic current stimulates NLRP3 to induce the release of both IL-1β and IL-18. Similar results were obtained with controls using the specific NLRP3 activator nigericin (*Figure 2B* and *Figure 2—figure supplement 1A*). Mechanistically, the use of an extracellular buffer with 40 mM of KCl decreased IL-1β release induced by nigericin and galvanic current application, but not the release of IL-1β induced by *Clostridium difficile* toxin B, which activates the Pyrin inflammasome, which is a K$^+$-efflux independent inflammasome (*Figure 2D*). However, two pulses of 12 mA of galvanic current for 6 s failed to decrease intracellular K$^+$ (*Figure 2E*, *Figure 2—figure supplement 1B*), but increasing the number of pulses of galvanic current to eight did result in a significant decrease of intracellular K$^+$ (*Figure 2E*). These data suggest that galvanic current slightly decreases intracellular K$^+$ when compared to the application of the K$^+$ ionophore nigericin (*Figure 2E*) and this explains the smaller concentration of IL-1β that is released by galvanic current compared to nigericin application (*Figure 2B*). After applying galvanic current we were able to detect the generation of the active p20 caspase-1 fragment, and processed IL-1β and GSDMD$^{NT}$ (*Figure 2F*). MCC950 was able to abrogate caspase-1 activation and the processed forms of IL-1β and GSDMD$^{NT}$ (*Figure 2F*), which suggests functional activation of caspase-1 and downstream signaling because of the activation of canonical NLRP3 and rules out the non-canonical NLRP3 activation that would result in GSDMD processing in the presence of MCC950. To ensure that

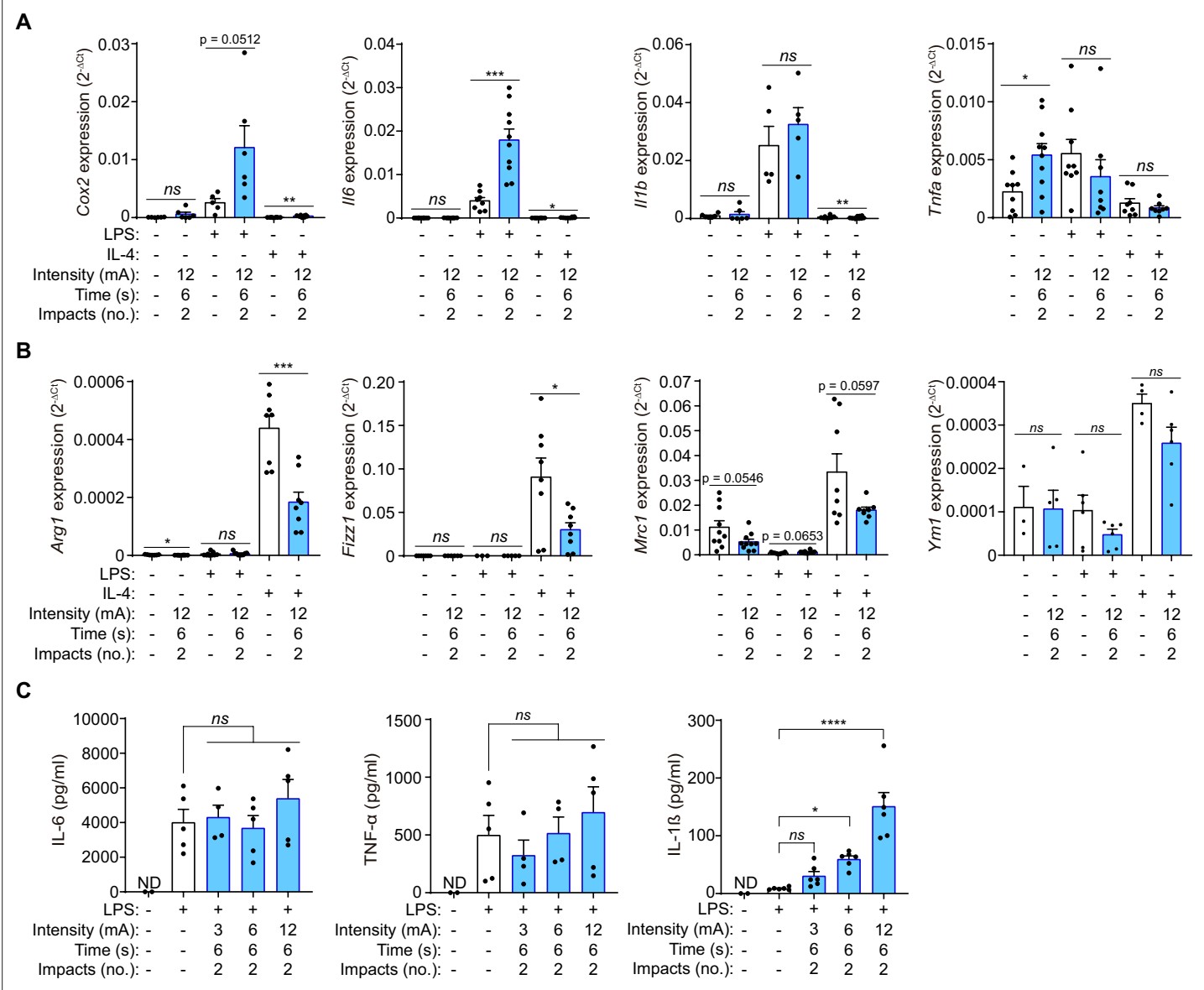

**Figure 1.** Galvanic current increases the M1 phenotype of macrophages. (**A**) Quantitative PCR for M1 genes *Cox2*, *Il6*, *Il1b* and *Tnfa* expression from mouse bone-marrow-derived macrophages (BMDMs) treated for 2 h with LPS (1 µg/ml) or 4 h with IL-4 (20 ng/µl) as indicated and then 2 impacts of 12 mA of galvanic current for 6 s. Cells were then further cultured for 6 h before analysis. Center values represent the mean and error bars represent s.e.m.; n = 5–10 samples of five independent experiments; for *Cox2* unpaired *t*-test except in the two first columns where Mann-Whitney tests were performed, for *Il6* Mann-Whitney test except in LPS *vs* LPS+ galvanic current comparison where unpaired *t*-tests were performed, for *Il1b* Mann-Whitney test, for *Tnfa* unpaired *t*-test except in LPS *vs* LPS+ galvanic current comparison where Mann-Whitney were performed, ***p < 0.0005, **p < 0.005, *p < 0.05, and *ns* p > 0.05. (**B**) Quantitative PCR for M2 genes *Arg1*, *Fizz1*, *Mrc1* and *Ym1* expression from BMDMs treated as in (**A**). Center values represent the mean and error bars represent s.e.m.; n = 3–10 samples of five independent experiments; unpaired *t*-test except LPS *vs* LPS+ galvanic current comparison where Mann-Whitney tests were performed, ***p < 0.0005, *p < 0.05, and *ns* p > 0.05. (**C**) IL-6, TNF-α and IL-1β release from BMDMs treated as in (**A**) but with different intensities of galvanic current (3, 6, 12 mA); ND: non detected. Center values represent the mean and error bars represent s.e.m.; n = 2 for untreated cells and n = 4–6 for treatment groups from four independent experiments; one-way ANOVA were performed comparing treated groups with the control group, ****p < 0.0001, *p < 0.05, and *ns* p > 0.05.

The online version of this article includes the following source data and figure supplement(s) for figure 1:

**Source data 1.** Raw data of panels of *Figure 1*.

**Figure supplement 1.** Device designed to apply galvanic current in 6-well plates.

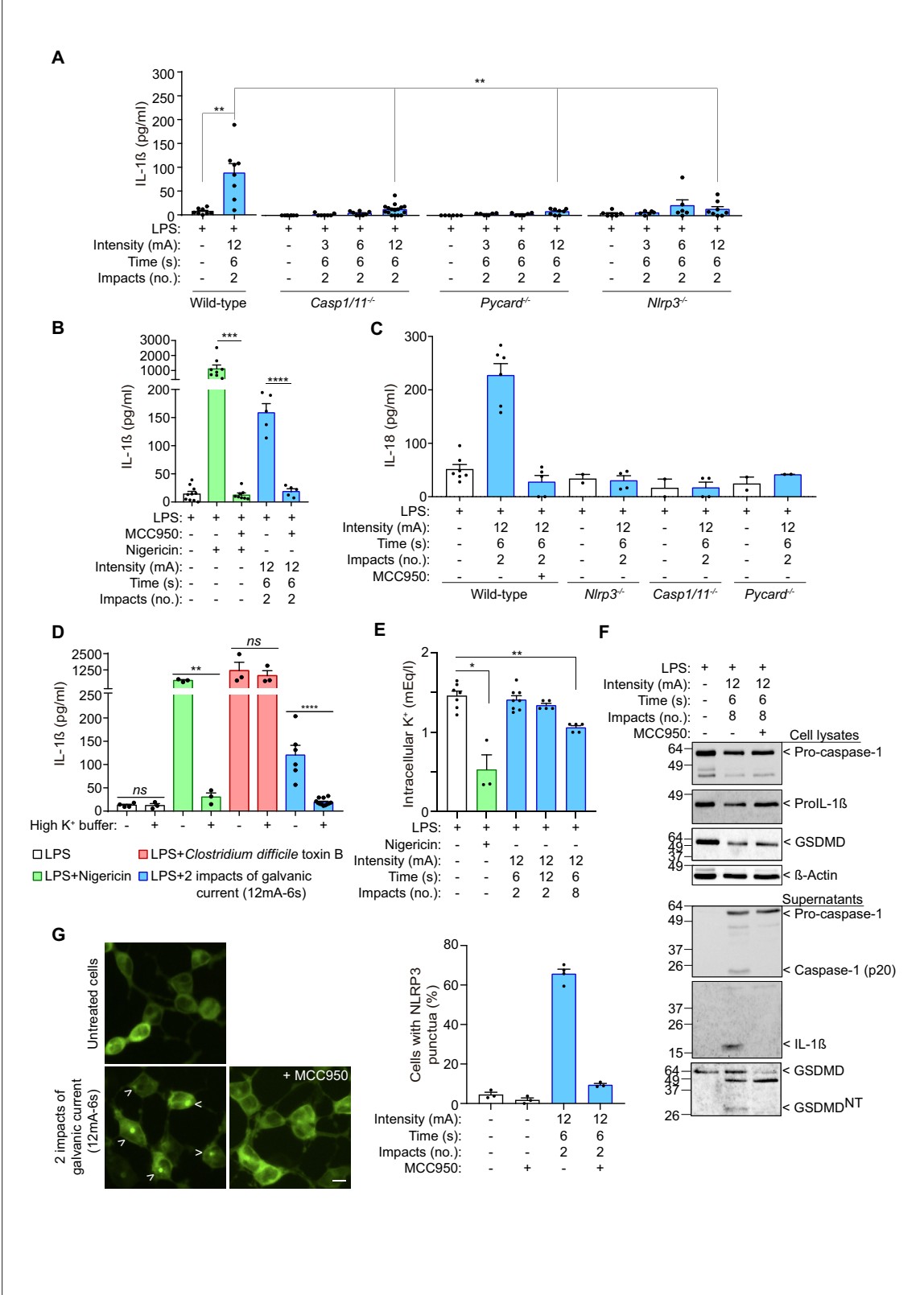

**Figure 2.** IL-1β release induced by galvanic current is dependent on the NLRP3 inflammasome. (**A**) IL-1β release from wild type, *Casp1/11⁻/⁻*, *Pycard⁻/⁻* and *Nlrp3⁻/⁻* mouse bone-marrow-derived macrophages (BMDMs) treated for 2 h with LPS (1 μg/ml) and then 2 impacts of different intensities of galvanic current (3, 6, 12 mA) for 6 s. Cells were then further cultured for 6 h before cytokine were measured in supernatant. Center values represent the mean and error bars represent s.e.m.; n = 6–16 samples of 10 independent experiments; LPS *vs* LPS+ galvanic current in wild type and wild type

*Figure 2 continued on next page*

*Figure 2 continued*

vs *Nlrp3*[-/-] unpaired *t*-test, wild type vs *Casp1/11*[-/-] and wild type vs *Pycard*[-/-] Mann-Whitney test, **p < 0.005. (**B**) IL-1β release from wild type BMDMs treated as in A but applying the NLRP3-specific inhibitor MCC950 (10 μM) 30 min before the galvanic current application and during the last 6 h of culture. As a control, cells were treated with nigericin (1.5 μM) rather than galvanic current. Center values represent the mean and error bars represent s.e.m.; n = 5–10 samples of 5 independent experiments; nigericin vs nigericin+ MCC950 Mann-Whitney test, galvanic current vs galvanic current+ MCC950 unpaired t-test, ****p < 0.0001 and ***p < 0.0005. (**C**) IL-18 release from BMDMs treated as in A. Center values represent the mean and error bars represent s.e.m.; n = 2–7 samples of at least two independent experiments. (**D**) IL-1β release from wild type BMDMs treated as in A but applying a buffer with 40 mM of KCl (high K[+] buffer) during the last 6 h of culture. As controls, cells were treated with nigericin (1.5 μM) or *Clostridium difficile* toxin B (1 μg/ml) rather than galvanic current. Center values represent the mean and error bars represent s.e.m.; n = 3–12 samples of four independent experiments; unpaired *t*-test, ****p < 0.0001, **p < 0.005 and *ns* p > 0.05. (**E**) Intracellular K[+] concentration from wild type BMDMs primed with LPS as in A, but then treated for 6 h with nigericin (1.5 μM) or two or 8 impacts of 12 mA for 6 or 12 s as indicated. Center values represent the mean and error bars represent s.e.m.; n = 3–8 samples of three independent experiments; Mann-Whitney test, **p < 0.005 and *p > 0.05. (**F**) Immunoblot of cell extract and supernatants for caspase-1, IL-1β, GSDMD and β-actin from wild type BMDMs treated as in B, but with eight impacts. Representative of n = 2 independent experiments. (**G**) Fluorescence microscopy of HEK293T cells stably expressing NLRP3–YFP after 6 h of application of 2 impacts of 12 mA of galvanic current for 6 s, the specific inhibitor MCC950 (10 μM) was added 30 min before the galvanic current; scale bar 10 μm; n = 3 independent experiments.

The online version of this article includes the following source data and figure supplement(s) for figure 2:

**Source data 1.** Uncropped western blots with the relevant bands boxed from *Figure 2D*.

**Source data 2.** Original files of the full raw unedited blots from *Figure 2D*.

**Source data 3.** Raw data of panels from *Figure 2*.

**Figure supplement 1.** Low number of galvanic current impacts does not induce a detectable decrease in intracellular K[+].

the galvanic current was activating NLRP3, we used HEK293T cells that were stably expressing NLRP3-YFP protein, a method widely used to assess NLRP3 activation (*Chen and Chen, 2018*; *Compan et al., 2012*; *Tapia-Abellán et al., 2021*; *Tapia-Abellán et al., 2019*). Fluorescence microscopy showed that 2 impacts of 12 mA of galvanic current for 6 s induced the formation of an intracellular NLRP3–YFP punctum, which was inhibited by the application of MCC950 (*Figure 2G*). All these data confirm that galvanic current activates the NLRP3 inflammasome, and since NLRP3-deficient macrophages failed to release IL-1β or IL-18, that no other inflammasome expressed in the macrophages was being activated.

## Galvanic current does not induce inflammasome-mediated pyroptosis

Given that GSDMD was processed and the N-terminus was detected after the application of galvanic current, we next assessed pyroptosis by means of Yo-Pro-1 uptake in cells to measure plasma membrane pore formation and cell viability, and LDH leakage from the cell to determine plasma membrane damage. Two impacts of galvanic currents of different intensities (3, 6, 12 mA) for a period of 6 s (conditions that induce IL-1β release as we show in *Figure 1C*) only induced a slight increase in cell death (*Figure 3A*). This increase in cell death was not associated with the activation of the inflammasome given that it was also present in macrophages deficient in NLRP3, ASC or caspase-1/11 (*Figure 3B*), which in turn suggests that it occurred independently of inflammasome-mediated pyroptosis. Increasing the number or the length of time of 12 mA impacts of galvanic current on macrophages, resulted in a time-dependent increase in cell death (*Figure 3A*), which correlates with higher concentrations of IL-1β and IL-18 release (*Figure 3C*). However, whereas IL-1β and IL-18 release was blocked by MCC950 (*Figure 3C*), LDH release was not dependent on NLRP3 activation (*Figure 3D*). This further corroborates the fact that the NLRP3 activation is dependent on the intensity and time of galvanic current application. Similarly, two impacts of 12 mA for a period of 6 s were unable to induce plasma membrane permeabilization measured in terms of Yo-Pro-1 uptake during a period of 3 h (*Figure 3E*). Yo-Pro uptake increased over 3 h in an intensity dependent manner (3, 6, 12 mA) when eight impacts were applied for 6 s (*Figure 3E*). This increase in plasma membrane permeabilization was not reversed after NLRP3 was blocked with MCC950 or when ASC-deficient macrophages were used (*Figure 3F*). All these results demonstrate that doses of 3 or 6 mA of galvanic current for impacts of 6 s do not compromise cell viability but are still able to induce an inflammatory response dependent on NLRP3 activation, in contrast to current intensities of 12 mA which, if prolonged cause significant cell death independently of the inflammasome.

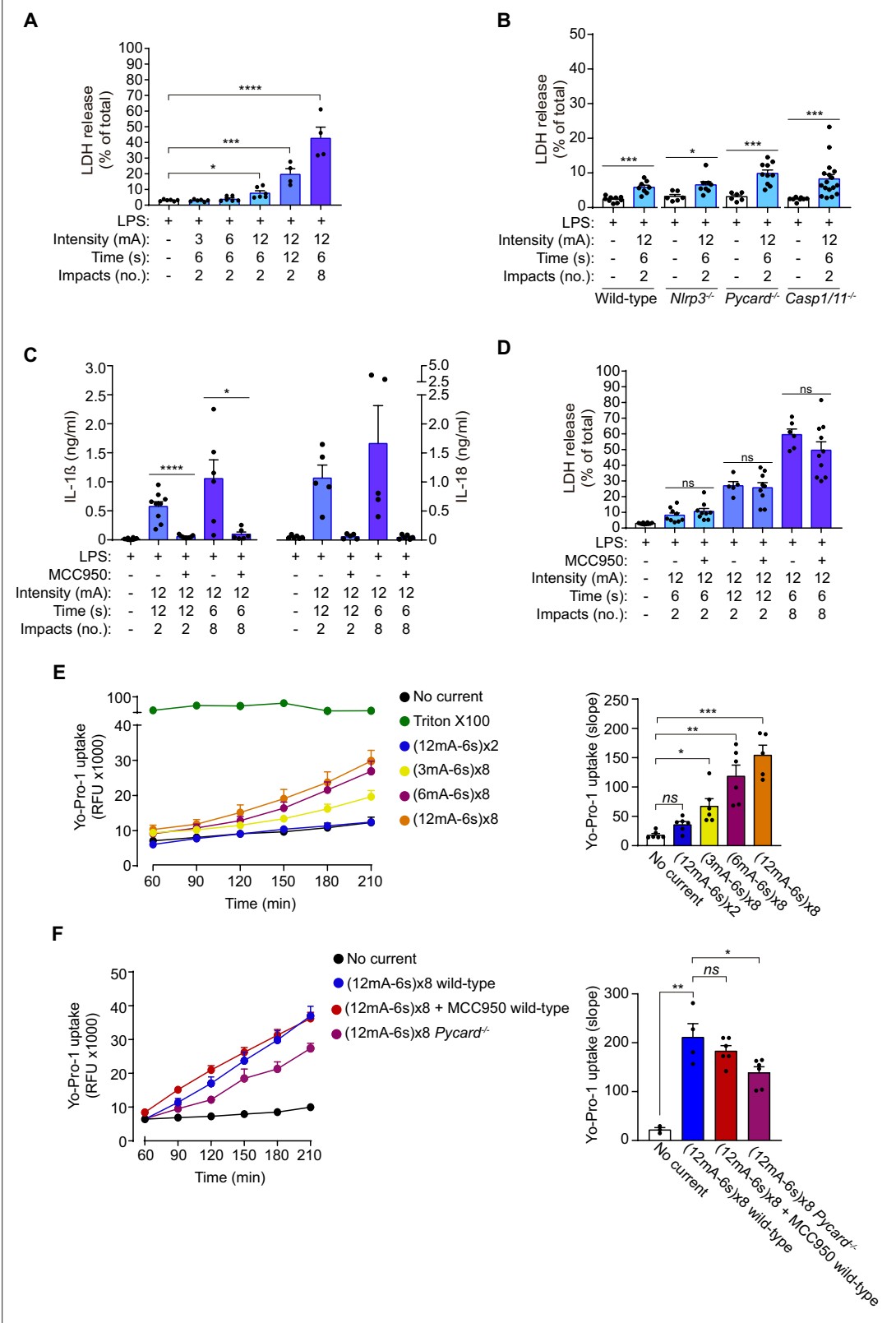

**Figure 3.** Galvanic current does not induce inflammasome-mediated pyroptosis. (**A**) Extracellular amount of LDH determining cell membrane damage from mouse bone-marrow-derived macrophages (BMDMs) treated for 2 h with LPS (1 µg/ml) and then two or 8 impacts of different intensities of galvanic current (3, 6, 12 mA) for 6 or 12 s were applied as indicated. Cells were then further cultured for 6 h before LDH determination in supernatant. Center values represent the mean and error bars represent s.e.m.; n = 3–4 samples of six independent experiments; Kruskal-Wallis test to compare LPS

*Figure 3 continued on next page*

*Figure 3 continued*

with increasing intensities of galvanic current, LPS *vs* LPS+ galvanic current (12mA-12s)x2 unpaired *t*-test, LPS *vs* LPS+ galvanic current (12mA-6s)x8, ****p < 0.0001, ***p < 0.0005 and *p < 0.05. (**B**) Extracellular amount of LDH from wild type, *Nlrp3*[-/-], *Pycard*[-/-], and *Casp1/11*[-/-] mouse BMDMs treated as in A. Center values represent the mean and error bars represent s.e.m.; n = 6–17 samples of 12 independent experiments; unpaired *t*-test except for *Casp1/11*[-/-] comparison, ***p < 0.0005 and *p < 0.05. (**C**) IL-1β (left) and IL-18 (right) release from wild type BMDMs treated as in A, but applying the NLRP3-specific inhibitor MCC950 (10 µM) during the last 6 h of culture. Center values represent the mean and error bars represent s.e.m.; n = 6–10 samples of 5 independent experiments; unpaired *t*-test, ****p < 0.0001 and *p < 0.005. (**D**) Extracellular amount of LDH from wild type BMDMs treated as in C. Center values represent the mean and error bars represent s.e.m.; n = 5–10 samples of five independent experiments; unpaired *t*-test, *ns* p > 0.005. (**E**) Kinetic of Yo-Pro-1 uptake (upper panel) or slope of the uptake (lower panel) in wild type BMDMs treated for 2 h with LPS (1 µg/ml) and then with different intensities of galvanic current (as indicated) or with the detergent triton X-100 (1 %) for 3.5 h. Center values represent the mean and error bars represent s.e.m.; n = 3–6 of three independent experiments; Kruskal-Wallis test, ***p < 0.0005, **p < 0.005 and *ns* p > 0.05. (**F**) Yo-Pro-1 uptake indicating plasma membrane pore formation and cell viability. Kinetic of Yo-Pro-1 uptake (upper panel) or slope of the uptake (lower panel) in wild type or *Pycard*[-/-] BMDMs treated as in E but, when indicated, the NLRP3 specific inhibitor MCC950 (10 µM) was added before galvanic current application. Center values represent the mean and error bars represent s.e.m.; n = 3–6 samples of three independent experiments; unpaired *t*-test, **p < 0.005, *p < 0.05 and *ns* p > 0.05.

The online version of this article includes the following source data for figure 3:

**Source data 1.** Raw data of panels from *Figure 3*.

## Galvanic current applied in tendon increases inflammation in vivo

When studying the effect of galvanic current in vivo, we found that application of 3 impacts of 3 mA of galvanic current for 3 s in the calcaneal tendon of mice increased the number of polymorphonuclear cells after 3 days when compared with tendons treated with needling alone (a puncture without current application, *Figure 4A and B*). This increase returned to basal after 7 days and stayed low up to 21 days after galvanic current application (*Figure 4B*). Similarly, the number of F4/80[+] macrophages increased 3 days after the application of galvanic current when compared to needling alone and returned to basal levels after 7 days (*Figure 4C and D*). Other immune cell types detected in the tendon, such as mastocytes, were not significantly increased by galvanic current when compared to needling alone (*Figure 4—figure supplement 1A*). Given that polymorphonuclear cells increased in a similar manner to macrophages, we then aimed to investigate if galvanic current might also induce the release of IL-1β from neutrophils. However, the application of galvanic current to LPS-primed neutrophils failed to release IL-1β or LDH (*Figure 4—figure supplement 1B*). Other histological features of the tendon (general structure of the tendon, number of tenocytes, shape and area of tenocyte nuclei or neo-vascularization) were also unaffected by the application of galvanic currents compared to needling alone (*Figure 4—figure supplement 1C-G*).

We next assessed the expression of different pro-inflammatory cytokines in the calcaneal tendon 3 days after 3 impacts of 3 mA of galvanic current application for 3 s to characterize the molecular inflammatory response in the tendon. Expression of *Il6*, *Il1a*, and *Il1b*, as well as the IL-1 receptor antagonist (*Il1rn*) and the chemokine *Cxcl10* all increased after percutaneous electrolysis when compared to needling alone (*Figure 5A*). However, as shown in *Figure 1* the increase in *Il6* cytokine gene expression induced by galvanic current in macrophages did not correlate with an increase in IL-6 cytokine secretion in vitro. Different NLRP3 inflammasome genes also exhibit an increase in expression (*Nlrp3*, *Pycard*, *Casp1*) when galvanic current was applied, but this increase was not significant when compared to needling (*Figure 5B*). *Gsdmd* expression was not upregulated in the tendons after galvanic current application (*Figure 5B*). These data suggest that galvanic current induces an inflammatory response driven by the infiltration of polymorphonuclear cells and macrophages, together with an increase in the expression of several cytokines and chemokines.

## The NLRP3 inflammasome controls the in vivo inflammatory response induced by galvanic current

In order to evaluate if the NLRP3 inflammasome mediates the inflammatory response in tendons after percutaneous electrolysis, we applied galvanic currents to the calcaneal tendon of *Nlrp3*[-/-] mice. Application of 3 impacts of 3 mA of galvanic current for 3 s in the calcaneal tendon of *Nlrp3*[-/-] mice resulted in a significant reduction of *Il1b*, *Il1rn* and *Cxcl10* expression after 3 days when compared to wild type mice (*Figure 6A*). Specific inflammasome associated genes, such as *Pycard*, *Casp1*, or *Gsdmd* (except for *Nlrp3*) did not affect their expression in the calcaneal tendon of *Nlrp3*[-/-] mice 3 days after the

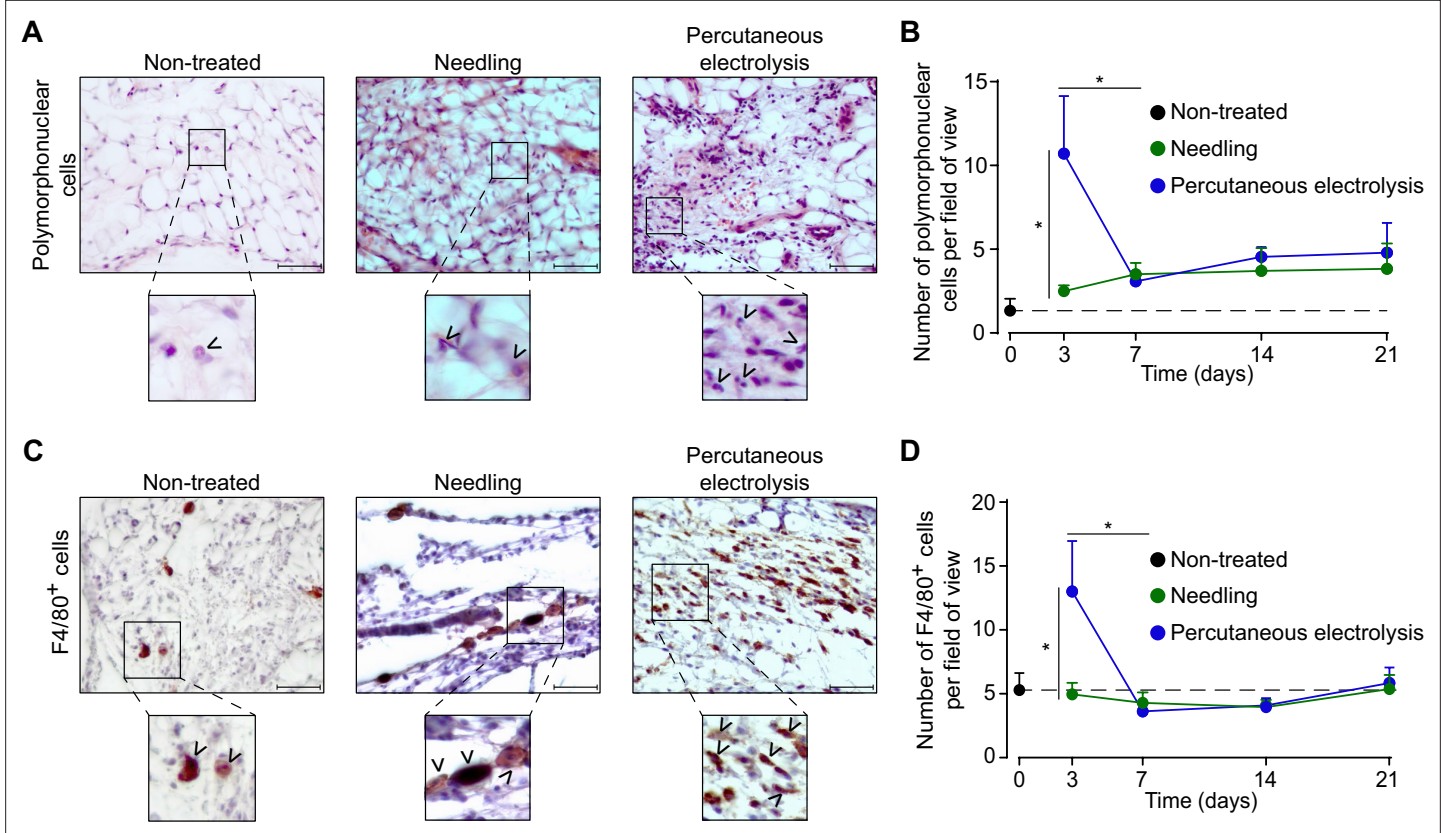

**Figure 4.** Galvanic current induces polymorphonuclear and macrophage infiltrate in the calcaneal tendon of mice. (**A**) Representative hematoxylin and eosin images of wild type mice calcaneal tendon after 3 days' application of 3 punctures with needle (needling, green) or 3 impacts of 3 mA for 3 s (blue). Scale bar: 50 μm. Magnification show the presence of polymorphonuclear cells (arrowheads). (**B**) Quantification of polymorphonuclear cells per field of view of calcaneal tendon sections treated and stained as described in A. Center values represent the mean and error bars represent s.e.m.; n = 7–8 independent animals; unpaired $t$-test, *p < 0.005. (**C**) Representative immunostaining images for the macrophage marker F4/80 from the calcaneal tendon of wild type mice treated as described in A. Scale bar: 50 μm. Magnification show the presence of F4/80-positive cells (arrowheads). (**D**) Quantification of F4/80-positive cells per field of view of calcaneal tendon sections treated and stained as described in C. Center values represent the mean and error bars represent s.e.m.; n = 8 independent animals; Mann-Whitney test, *p < 0.005.

The online version of this article includes the following source data and figure supplement(s) for figure 4:

**Source data 1.** Raw data of panels from *Figure 4C and D*.

**Source data 2.** Representation of raw data of panels from *Figure 4C and D*.

**Figure supplement 1.** Galvanic current does not affect tendon mastocytes, tenocytes, or vascularity.

application of galvanic current when compared to wild type mice (*Figure 6B*). Surprisingly, galvanic current tended to increase both the expression of *Il6* in the tendons of *Nlrp3*[-/-] after 3 days (*Figure 6C*) and the number of polymorphonuclear cells (*Figure 6D*). However, the number of macrophages was not affected in the *Nlrp3*[-/-] calcaneal tendon when galvanic current was applied (*Figure 6D*). We also confirmed a decrease in *Il1b* and *Cxcl10* expression in the tendons of *Pycard*[-/-] mice 3 days after the application of galvanic current (*Figure 6—figure supplement 1A, B*), which suggests that the NLRP3 inflammasome is important in modulating part of the inflammatory response after the application of galvanic current.

## The NLRP3 inflammasome induces a tissue regenerative response to galvanic current application that increases tendon stiffness

Galvanic current application has been widely used to resolve chronic tendinopathies in different tendons (*Abat et al., 2016*; *Rodríguez-Huguet et al., 2020*; *Valera-Garrido et al., 2014*), and as an example we present here a case where lateral epicondylitis was resolved in 6 weeks after four sessions of percutaneous electrolysis through the application of 3 mA of galvanic current for 3 periods

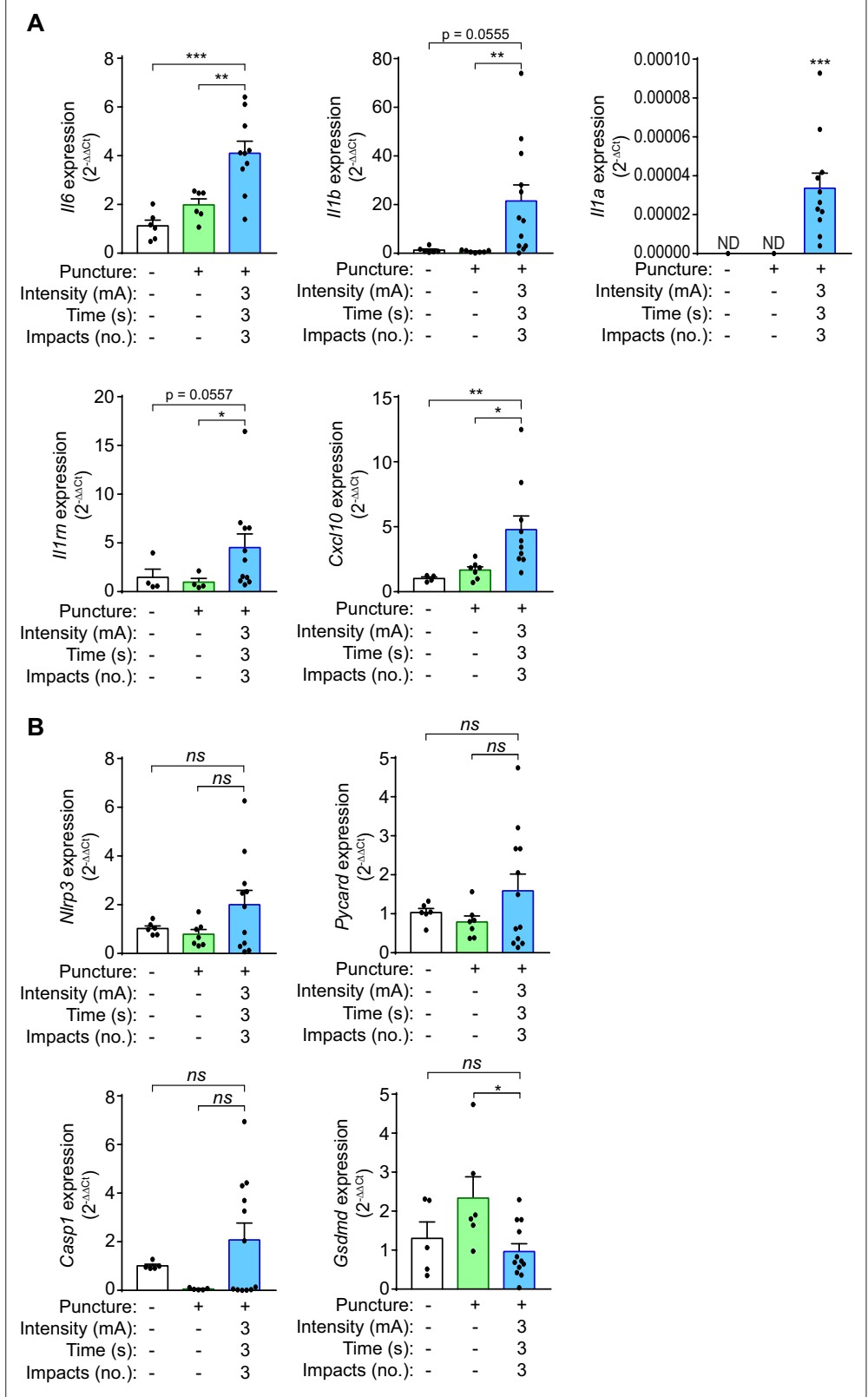

**Figure 5.** Galvanic current induces proinflammatory cytokine expression in the calcaneal tendon of mice.
(**A,B**) Quantitative PCR for the indicated genes normalized to *Actb* in the calcaneal tendon of wild type mice after
3 days of applying three punctures with needle (needling, green) or 3 impacts of 3 mA for 3 s (blue), and compared
to the expression of genes in non-treated tendons. Center values represent the mean and error bars represent

*Figure 5 continued on next page*

*Figure 5 continued*

s.e.m.; n = 4–12 independent animals; for *Il6*, *Nlrp3*, *Pycard,* and *Gsdmd* unpaired *t*-test, for *Il1b* untreated *vs* galvanic current Mann-Whitney and puncture *vs* galvanic current unpaired *t*-test, for *Cxcl10*, *Il1rn* and *Casp1/11* Mann-Whitney test, for *Il1a* one sample Wilcoxon test (ND: non detected), \*\*\*p < 0.0005, \*\*p < 0.005, \*p < 0.05 and *ns* p > 0.05.

The online version of this article includes the following source data for figure 5:

**Source data 1.** Raw data of panels from *Figure 5*.

of 3 seconds (3:3:3) (*Figure 7—figure supplement 1A*), in accordance with the clinical protocol previously described by Valera-Garrido and Minaya-Muñoz (*Valera-Garrido and Minaya-Muñoz, 2016*). During tissue regeneration, the production of new extracellular matrix through collagen deposition is a key process (*Shook et al., 2018*; *Wynn, 2008*). In order to determine if the inflammatory response mediated by the NLRP3 inflammasome after galvanic current application is important for tissue regeneration, we measured *Tgfb1* expression as a key factor in inducing collagen production. We found that in vivo the expression of *Tgfb1* 3 days after the application of galvanic current in the calcaneal tendon of mice was dependent on NLRP3 (*Figure 7A*). Likewise, after 7 days of percutaneous electrolysis the levels of type III collagen were decreased, with a parallel increase in type I collagen when compared to needling alone (*Figure 7B*, *Figure 7—figure supplement 1D*). However, percutaneous electrolysis did not affect collagen fiber properties (width or length) when compared to needling alone (*Figure 7—figure supplement 1B,C*). The increase in type I collagen 7 days after galvanic current application was reduced in *Nlrp3*[-/-] mice (*Figure 7C*). NLRP3 also controlled the structural dispersion of collagen fibers, which was decreased by galvanic current application in wild type mice, but increased in *Nlrp3*[-/-] mice (*Figure 7D*, *Figure 7—figure supplement 1E*). All these results suggest that the NLRP3 inflammasome controls the response of galvanic current thus inducing type I collagen production and arranging the collagen fibers. This controlled deposition of collagen fibers induced by galvanic current increased tendon stiffness and decreased the maximum tension supported by the tendon (*Figure 7E*). NLRP3 inflammasome was responsible for increased tendon stiffness after galvanic current application (*Figure 7E*). Overall, we found that the application of galvanic current is able to activate the NLRP3 inflammasome and induce the release of IL-1β, thus initiating an inflammatory response that regenerates the tendon by increasing type I collagen, arranging the collagen fibers and increasing the resistance of the tendon to changes in length (*Figure 7—figure supplement 2*).

## Discussion

In this study, we demonstrate how the application of galvanic current induces in macrophages a pro-inflammatory signature, mainly characterized by the activation of the NLRP3 inflammasome and the release of mature IL-1β and IL-18. This is in agreement with the fact that the NLRP3 inflammasome is a key pathway for controlling inflammation in the absence of pathogenic microorganisms (in sterile conditions) by executing a pro-inflammatory type of cell death called pyroptosis (*Broz et al., 2020*; *Broz and Dixit, 2016*; *Liston and Masters, 2017*). Here, we found that galvanic current application, a technique that has been widely used to resolve chronic lesions in humans (*Valera-Garrido et al., 2014*), was able to activate the NLRP3 inflammasome and induce IL-1β and IL-18 release, but with very little associated pyroptotic cell death. This could be due to two potentially different mechanisms: (*i*) that after galvanic current application an alternative means of processing GSDMD occurs which is independent of NLRP3 and which could inactivate its N-terminal lytic domain, as has been previously found for caspase-3 processing GSDMD (*Taabazuing et al., 2017*); and/or (*ii*) that the small amounts of GSDMD[NT] found could create a small number of pores in the plasma membrane, thus facilitating their repair by the endosomal sorting complexes required for transport machinery, which in turn leads to an hyperactive state of the macrophage (*Evavold et al., 2018*; *Rühl et al., 2018*). During this state of the macrophage, IL-1β is released in the absence of cell death (*Evavold et al., 2018*). However, an increase in the intensity and the time of galvanic current application leads to an increase in cell death that was independent of the inflammasome and could be related to the technique per se. Therefore, clinical application of current intensities above 6 mA would probably lead to necrosis of the tissue and not to an efficient reparative process. Galvanic currents of 3 and 6 mA for 2 impacts of 6 s are both able to induce NLRP3 inflammasome activation in vitro and lead to phenotypic changes in the tendon

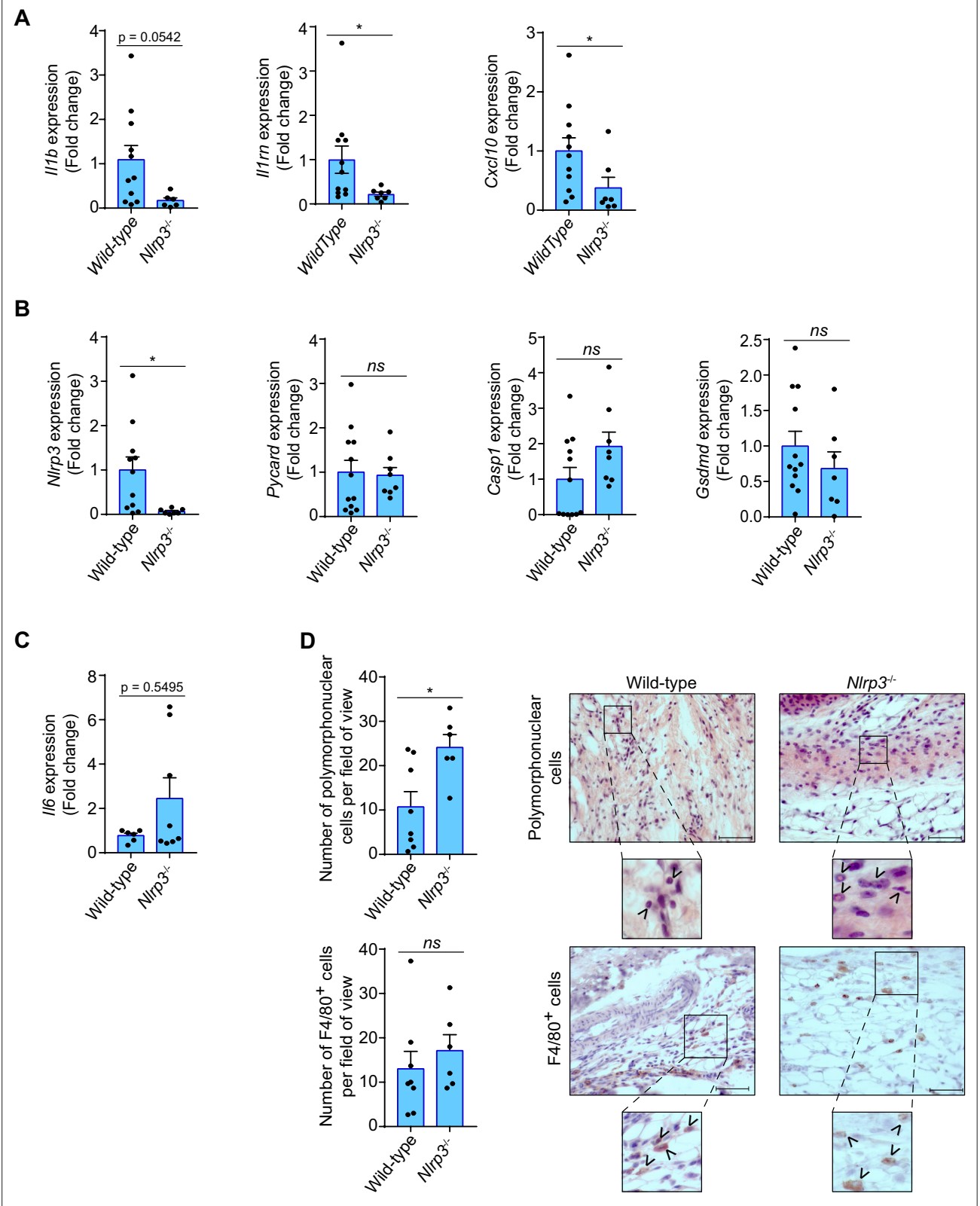

**Figure 6.** Inflammatory response in the calcaneal tendon of *Nlrp3*$^{-/-}$ mice after galvanic current application. (**A–C**) Quantitative PCR for the indicated genes in the calcaneal tendons of *Nlrp3*$^{-/-}$ mice (calculated as $2^{-\Delta\Delta Ct}$) normalized to the expression in wild type (calculated as $2^{-\Delta Ct}$) after 3 days of 3 impacts of 3 mA for 3 s. Center values represent the mean and error bars represent s.e.m.; n = 3–12 independent animals; for *Il1b*, *Nlrp3*, *Pycard*, *Casp1/11*, and *Gsdmd* unpaired *t*-test, for *Il1rn* and *Cxcl10* Mann-whitney test, *p < 0.05 and *ns* p > 0.05. (**D**) Quantification of polymorphonuclear (top)

*Figure 6 continued on next page*

*Figure 6 continued*

and F4/80 positive cells (bottom) per field of view from wild type and *Nlrp3*⁻/⁻ mice calcaneal tendon treated as in A. Center values represent the mean and error bars represent s.e.m.; n = 6–8 independent animals; unpaired *t*-test, *p < 0.05 and *ns* p > 0.05. Representative hematoxylin and eosin images (top) and F4/80 immunostaining (bottom) of calcaneal tendon quantified. Scale bar: 50 μm. Magnification shows the presence of polymorphonuclear (top) or F4/80 cells (bottom) denoted by arrowheads.

The online version of this article includes the following source data and figure supplement(s) for figure 6:

**Source data 1.** Raw data of panels from *Figure 6*.

**Figure supplement 1.** Cytokine expression in the calcaneal tendon of *Pycard*⁻/⁻ mice after galvanic current application.

in vivo. This is in line with the fact that 3 mA of galvanic current is able to induce a clinically significant regeneration of lesions (*García-Vidal et al., 2019*; *Margalef, 2019*; *Medina-Mirapeix et al., 2019*; *Valera-Garrido et al., 2014*). High-intensity doses for long periods of time or repeated impacts could induce massive tissue necrosis and are therefore not recommended for clinical practice.

The activation of the NLRP3 inflammasome induced by galvanic currents was found to be dependent on K⁺ efflux, because extracellular high concentrations of K⁺ were able to block IL-1β release, and high intensities of galvanic current decreased intracellular K⁺. This is similar to the effect of the well-studied K⁺ ionophore nigericin, which dramatically decreases intracellular K⁺ and releases IL-1β (*Munoz-Planillo et al., 2013*; *Pétrilli et al., 2007*; *Próchnicki et al., 2016*). In fact, the amount of IL-1β released from galvanic current activated macrophages was lower than when macrophages were activated with nigericin, which suggests that the NLRP3 activation is correlated with the decrease in intracellular K⁺ (*Tapia-Abellán et al., 2021*). The low NLRP3 activation induced by galvanic current application could result in a moderate inflammatory response in vivo that is beneficial for tissue regeneration. In fact, NLRP3 was important for inducing an inflammatory response in vivo with increased quantities of different cytokines including *Il1b* or *Cxcl10*, that conditioned the structure and functions of the treated tendons. However, NLRP3 deficiency does not affect *Il6* production or the infiltration of polymorphonuclear cells when galvanic currents are applied in vivo. This shows that NLRP3 induced by galvanic current is able to control a specific inflammatory program in vivo, but it probably do not affect the IL-6-mediated infiltration of polymorphonuclear cells in tendons.

Exacerbated NLRP3 activation could led to fibrosis (*Alegre et al., 2017*; *Gaul et al., 2021*), which suggests that NLRP3 could control collagen deposition. The mild activation of NLRP3 found after galvanic current application was associated with increased production of *Tgfb1*, increased in collagen type I vs type III in tendons, and reduced structural dispersion of collagen fibers. This is associated with an increase in tendon stiffness, which is related to the resistance of the tendon to changes in length. Tendon stiffness is reduced during ageing and results in weaker tendons (*Krupenevich et al., 2021*). This could explain previous clinical findings related to galvanic current therapy, such as the fact that almost all patients treated with galvanic current in tendon lesions had no relapses at long term (*Valera-Garrido et al., 2014*) and the fact that application of galvanic current in human tendon lesions in combination with exercise therapy achieved greater functional recovery compared to exercise alone (*Moreno et al., 2017*), and also compared to the combination of exercise with other rehabilitation interventions such as electrotherapy or mechanical intervention (*Abat et al., 2016*; *Rodríguez-Huguet et al., 2020*). Thus, our findings could help professionals to choose and combine rehabilitative and orthopedic treatments. However, a limitation of our study is that using animals restricts us to employing the ultrasound-guided puncture when applying galvanic current to the calcaneal tendon of mice, because it is the largest accessible tendon, whereas in humans galvanic current has been applied supraspinatus (*Rodríguez-Huguet et al., 2020*), patellar (*Abat et al., 2016*) and lateral epicondyle (*Valera-Garrido et al., 2014*) tendons. In all tendons, applying galvanic current was found to have a similar clinical benefit, but we cannot rule out the fact that the calcaneal tendon of mice would present a different response to galvanic current. In fact, different species have been reported to have different inflammatory tenocyte responses (*Oreff et al., 2021*), and although mice and humans present high overall similarities in their tenocyte responses (*Oreff et al., 2021*), it is necessary to apply galvanic current in the presence of specific NLRP3 blockers, such as MCC950, in additional animal models with species other than mice.

Therefore, this study reports how galvanic current is a feasible technique applied in vivo to activate the NLRP3 inflammasome and induce a local inflammatory response to enhance a collagen-mediated

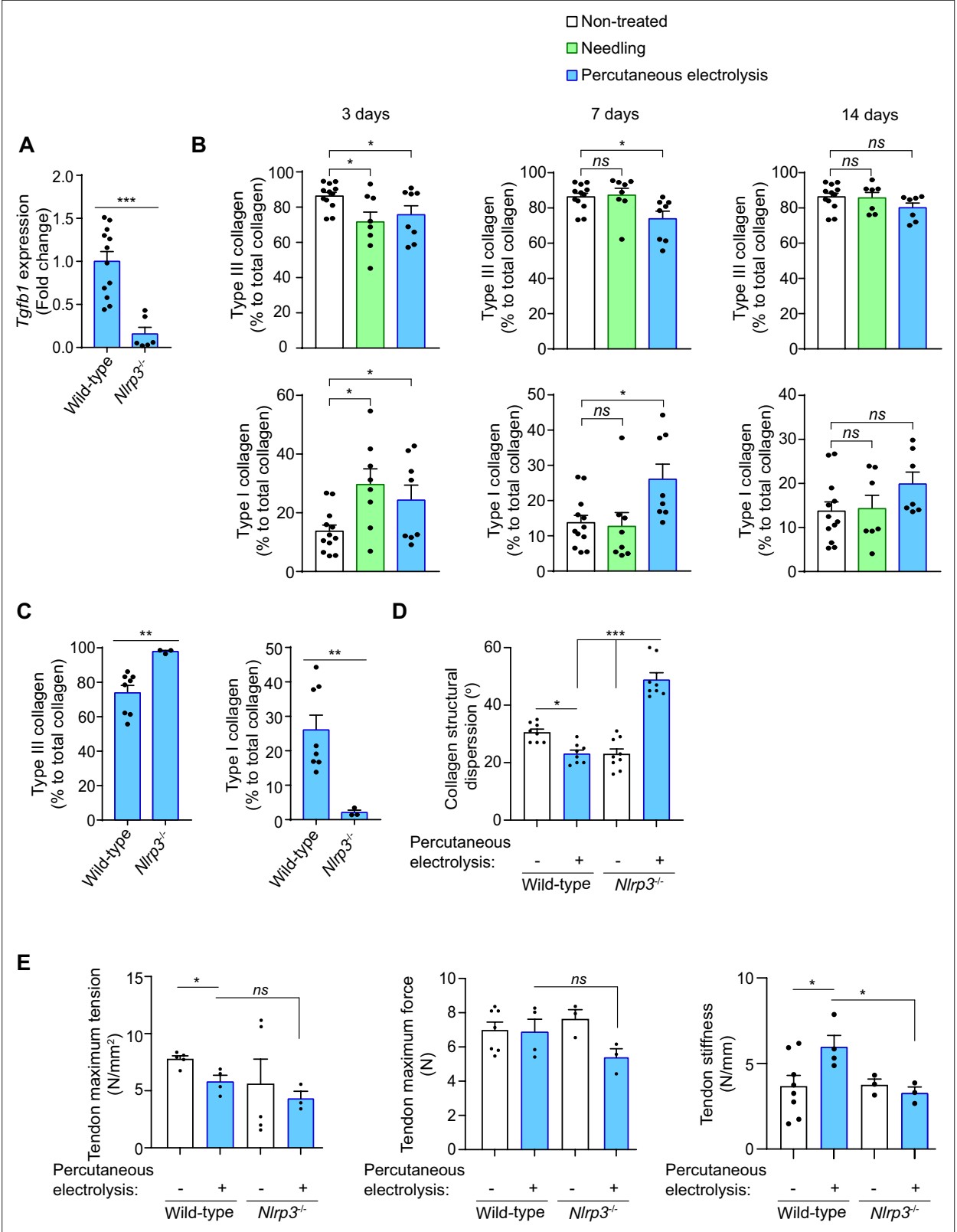

**Figure 7.** Galvanic current increase in type I collagen via NLRP3 inflammasome. (**A**) Quantitative PCR for *Tgfb1* in the calcaneal tendons of *Nlrp3*[-/-] mice (calculated as 2[-ΔΔCt]) normalized to the expression in wild type (calculated as 2[-ΔΔCt]) after 3 days of 3 impacts of 3 mA for 3 s. Center values represent the mean and error bars represent s.e.m.; n = 6–12 independent animals; Mann-Whitney test, ***P < 0.0005. (**B,C**) Quantification of the collagen type I and III in calcaneal tendon sections stained with picrosirius red from wild type (**B,C**) and *Nlrp3*[-/-] (**C**) mice after 3, 7 or 14 days (**B**) or 3 days (**C**) application of

*Figure 7 continued on next page*

*Figure 7 continued*

punctures with needle (needling, green) or 3 impacts of 3 mA for 3 s (blue), or in non-treated tendons (white). Center values represent the mean and error bars represent s.e.m.; n = 3–12 independent animals; for 3 days and panel C unpaired *t*-test, for 7 days non-treated *vs* needling Mann-Whitney test and non-treated *vs* percutaneous electrolysis unpaired *t*-test, for 14 days untreated *vs* needling unpaired *t*-test and untreated *vs* percutaneous electrolysis Mann-Whitney test, \*\*p < 0.005, \*p < 0.05 and *ns* p > 0.05. (**D**) Quantification of collagen structural dispersion in calcaneal tendon sections imaged with second harmonic generation microscopy and calculated with an algorithm based on the Hough transform from wild type and *Nlrp3*$^{-/-}$ mice after 7 days of 3 impacts of 3 mA for 3 s (blue), or in non-treated tendons (white). Center values represent the mean and error bars represent s.e.m.; n = 3 independent animals imaged at two or three different tendon areas, one-way ANOVA \*\*\*p < 0.0005 and \*p < 0.05. (**E**) Biomechanical testing of calcaneal tendon from wild type and *Nlrp3*$^{-/-}$ mice after 14 days of 3 impacts of 3 mA for 3 s (blue) or from non-treated tendons (white). Center values represent the mean and error bars represent s.e.m.; n = 3–8 independent animals; unpaired *t*-test, \*p < 0.05 and *ns* p > 0.05.

The online version of this article includes the following source data and figure supplement(s) for figure 7:

**Source data 1.** Raw data of panels from *Figure 7*.

**Figure supplement 1.** Galvanic current do not change properties of the collagen.

**Figure supplement 2.** Model summarizing the action of galvanic current in tendon regeneration.

**Figure supplement 3.** Clamps designed to measure tendon tension.

**Figure supplement 4.** Accumulation matrix of the algorithm based on the Hough transform from collagen fibers imaged with second harmonic generation microscopy of calcaneal tendons from wild type and *Nlrp3*$^{-/-}$ mice after 7 days of 3 impacts of 3 mA for 3 s.

regeneration process in the tendon, establishing the molecular mechanism of percutaneous electrolysis for the treatment of chronic lesions.

# Materials and methods

## Key resources table

| Reagent type (species) or resource | Designation | Source or reference | Identifiers | Additional information |
|---|---|---|---|---|
| Genetic reagent (*Mus musculus*, male) | NLRP3-deficient mice (*Nlrp3*$^{-/-}$) - C57 BL/6 Nlrp3$^{tm1Vmd}$ | Jackson laboratories | RRID:MGI:5468973 | In vivo mouse models and biological samples. |
| Genetic reagent (*Mus musculus*, male) | ASC-deficient mice (*Pycard*$^{-/-}$) - C57 BL/6 Pycard$^{tm1Vmd}$ | Jackson laboratories | RRID:MGI:3047277 | In vivo mouse models and biological samples. |
| Genetic reagent (*Mus musculus*, male) | Casp1/11-deficient mice (*Casp1/11*$^{-/-}$) - B6N.129S2-Casp1$^{tm1Flv}$ | Jackson laboratories | RRID:MGI:5467384 | Biological samples. |
| Antibody | Anti-GSDMD (rabbit monoclonal, clone EPR19828) | Abcam | Cat#: ab209845, RRID:AB_2783550 | WB (1:2000) |
| Antibody | Anti-Caspase 1 p20 (mose monoclonal, clone Casper-1) | Adipogen | casper-1, Cat# AG-20B-0042, RRID:AB_2490248 | WB (1:1000) |
| Antibody | Anti-IL-1β (rabbit polyclonal) | Santa Cruz | H-153 Cat#: SC-7884, RRID:AB_2124476 | WB (1:1000) |
| Sequence-based reagent | KiCqStart SYBR Green Primers | Sigma-Aldrich | *Tnfa* (NM_013693) *Il6* (NM_031168) *Il1b* (NM_008361) *Actb* (NM_007393) *Cox2 (Ptgs2)* (NM_011198) *Arg1* (NM_007482) *Fizz1 (Retnlb)* (NM_023881) *Mrc1* (NM_ 008625) *Ym1 (Chil1)* (NM_007695001080219) | qRT-PCR |

*Continued on next page*

*Continued*

| Reagent type (species) or resource | Designation | Source or reference | Identifiers | Additional information |
|---|---|---|---|---|
| Commercial assay or kit | ELISA for IL-1β | R&D SystemsThermo Fisher Scientific | Cat# 88-7013-88, RRID:AB_257494688-7013-88xxx | ELISA for cell supernatant |
| Commercial assay or kit | ELISA for TNFα | Thermo Fisher Scientific | Cat# 88-7324-88, RRID:AB_257508088-7324-88xxx | ELISA for cell supernatant |
| Commercial assay or kit | ELISA for IL-6 | R&D Systems | Cat# M6000B, RRID:AB_2877063 M6000Bxxx | ELISA for cell supernatant |
| Commercial assay or kit | ELISA for IL-18 | Thermo Fisher Scientific | Cat# BMS618/3, RRID:AB_2575692BMS618-3xxx | ELISA for cell supernatant |
| Chemical compound, drug | MCC950 | Sigma-Aldrich | PZ0280 | Cell culture: 10 µM |

## Animals and percutaneous needle puncture procedure

All experimental protocols for animal handling were refined and approved by the local animal research ethical committee (references 241/2016 and 541/2019) and the Animal Health Service of the General Directorate of Fishing and Farming of the Council of Murcia (*Servicio de Sanidad Animal, Dirección General de Ganadería y Pesca, Consejería de Agricultura y Agua de la Región de Murcia,* reference A13160702). C57/BL6 mice (wild type) were obtained from Jackson Laboratories. NLRP3-deficient mice (*Nlrp3*$^{-/-}$) and Caspase-1/11-deficient mice (*Casp-1/11*$^{-/-}$) in C57/BL6 background were a generous gift of I. Coullin. For all experiments, mice between 8 and 10 weeks of age were used. Mice were bred in specific pathogen-free conditions with a 12:12 h light-dark cycle and used in accordance with the animal experimentation guidelines of the *Hospital Clínico Universitario Vírgen de la Arrixaca,* and the Spanish national (RD 1201/2005 and Law 32/2007) and EU (86/609/EEC and 2010/63/EU) legislation. Percutaneous needle puncture was performed with 0.2 x 16 mm needles (Agupunt) in the calcaneal tendon in isoflurane (Zoetis) anesthetized mice; galvanic current was applied using Physio Invasiva equipment (Prim) delivering three impacts of 3 mA for 3 s and compared to a puncture without current application. Paws without puncture were also used as controls. 3, 7, 14, and 21 days after puncture, animals were euthanized and paws were collected for histopathology or gene expression. Only calcaneal tendon was dissected for gene expression and the zone between gastrocnemius and calcaneus, including tendon, adipose tissue, tibia, and peroneus, was dissected for histopathology.

## Patient

A male patient, 36 years old with lateral epicondylalgia in the right elbow for 6 months, with pain and functional impairment. Resistant to conventional treatments (physiotherapy, oral non-steroidal anti-inflammatory and local corticoid infiltrations). Ultrasound analysis showed extensor joint tendon degeneration correlating with positive orthopedic tests. The patient was subjected to four sessions of percutaneous electrolysis with an intensity of 3 mA of galvanic current for 3 s, 3 times (3:3:3), according to the protocol by *Valera-Garrido and Minaya-Muñoz, 2016*.

## Cell culture and treatments

Bone-marrow-derived macrophages (BMDMs) were obtained from wild type, *Casp1/11*$^{-/-}$, *Nlrp3*$^{-/-}$ and *Pycard*$^{-/-}$ mice. Cells were differentiated for 7 days in DMEM (Lonza) supplemented with 25% L929 medium, 15% fetal bovine serum (FCS, Life Technologies), 100 U/ml penicillin/streptomycin (Lonza), and 1% L-glutamine (Lonza). After differentiation, cells were primed for 2 h with 1 µg/ml *E. coli* lipopolysaccharide (LPS) serotype O55:B5 at (Sigma-Aldrich) or for 4 h with 20 ng/ml recombinant mouse IL-4 (BD Pharmigen). Cells were then washed twice with isotonic buffer composed of 147 mM NaCl, 10 mM HEPES, 13 mM glucose, 2 mM CaCl$_2$, 1 mM MgCl$_2$, and 2 mM KCl, pH 7.4, and then treated in OptiMEM (Lonza) with different intensities and time of galvanic current (as indicated in the text and Figure legends) using an ad hoc adaptor for six well plates (*Figure 1—figure supplement 1*), and then cultured for 6 h. Alternatively and as a positive control, after LPS-priming macrophages were treated for 6 h in OptiMEM with 1.5 µM nigericin (Sigma-Aldrich) or 1 µg/ml *Clostridium difficile* toxin B (Enzo Life Sciences) to activate NLRP3 and Pyrin inflammasomes respectively. In some experiments, cells

were treated with 10 µM of the NLRP3 inflammasome inhibitor MCC950 (CP-456773, Sigma-Aldrich) after LPS priming and during inflammasome activation.

Stable HEK293T cell lines constitutively expressing NLRP3-YFP have already been produced (*Tapia-Abellán et al., 2019*). The cell line was not authenticated, but was routinely tested with the Myco-Probe Mycoplasma Detection Kit following manufacturer instructions (R&D Systems) to ensure that it was free of mycoplasma. Cells were subjected to two pulses of 12 mA of galvanic current during 6 s each and incubated for 6 h at 37 °C and 5% $CO_2$ before being analyzed by fluorescence microscopy.

## Fluorescence microcopy

Stable HEK293T cells were imaged with a Nikon Eclipse Ti microscope equipped with a 20× S Plan Fluor objective (numerical aperture, 0.45), a digital Sight DS-QiMc camera (Nikon), 472 nm/520 nm, 543 nm/593 nm filter sets (Semrock), and the NIS-Elements AR software (Nikon). Images were analyzed with ImageJ software (NIH).

## LDH release, Yo-Pro uptake assay and K⁺ measurements

The presence of lactate dehydrogenase (LDH) in cell culture supernatants was measured using the Cytotoxicity Detection kit (Roche), following manufacturer's instructions. It was expressed as the percentage of the total amount of LDH present in the cells. For Yo-Pro uptake, macrophages were preincubated for 5 min at 37 °C with 2.5 µM of Yo-Pro-1 iodide (Life Technologies) after galvanic current application or 1% triton X100 (Sigma-Aldrich) application. Yo-Pro-1 fluorescence was measured after the treatments every 5 min for the first 30 min and then every 30 min for the following 3 h with an excitation wavelength of 478 ± 20 nm and emission of 519 ± 20 nm in a Synergy neo2 multi-mode plate reader (BioTek). Intracellular K⁺ was quantified using macrophages lysates, as has already been reported (*Compan et al., 2012*), and measured by indirect potentiometry on a Cobas 6000 with ISE module (Roche).

## Western blot and ELISA

After cell stimulation, cells extracts were prepared in cold lysis buffer and incubated at 4 °C for 30 min and then centrifuged at 12,856 x*g* for 10 min at 4 °C. Cells supernatants were centrifuged at 12,856 x*g* for 30 s at 4 °C and concentrated by centrifugation at 11,000 x*g* for 30 min at 4 °C through a column with a 10 kDa cut-off (Merk-Millipore). Cell lysates and concentrated supernatants were mixed with loading buffer (Sigma), boiled at 95 °C for 5 min, resolved in 15% polyacrylamide gels and transferred to nitrocellulose membranes (BioRad). Different primary antibodies were used for the detection of interest proteins: anti-IL-1β rabbit polyclonal (1:1000, H-153, SC-7884, Santa Cruz), anti-caspase-1 (p20) mouse monoclonal (1:1000, casper-1, AG-20B-0042, Adipogen), anti-gasdermin D rabbit monoclonal (1:2000, EPR19828, ab209845, Abcam) and anti-β-Actin mouse monoclonal (1:10,000, Santa Cruz). Appropriate secondary antibody conjugated with HRP was used at 1:5000 dilution (Sigma) and developed with ECL plus (Amhershan Biosciences) in a ChemiDoc HDR (BioRad). Uncropped western blots are shown in *Figure 2—source data 1 and 2*. The concentration of IL-1β, IL-18, TNF-α, and IL-6 in cell supernatants was determined by ELISA following the manufacturer's instructions. ELISA were purchased from R&D Systems or Thermo Fisher Scientific (see Key Resources Table). Results were read in a Synergy Mx plate reader (BioTek).

## Quantitative reverse transcriptase-polymerase chain reaction (RT-PCR) analysis

Total RNA extraction was performed using macrophages or mice tendons dissected as described above. Macrophage total RNA extraction was performed using the RNeasy Mini Kit (Qiagen) following the manufacturer's instructions. Total RNA extraction from mice tendons was performed using Qiazol lysis reagent (Qiagen) and samples were homogenized using an Omni THQ homogenizer. After homogenization, samples were incubated for 5 min at room temperature and centrifuged at 12,000 x*g* for 15 min at 4 °C. After centrifugation, the upper phase was collected and one volume of 70% ethanol was added. Samples were loaded in RNeasey Mini Kit columns and total RNA isolation was performed following manufacturer's instructions. In both cases a step with a treatment with 10 U/µl DNase I (Qiagen) was added during 30 min. Reverse transcription was performed using iScript cDNA Synthesis kit (BioRad). The mix SYBR Green Premix ExTaq (Takara) was used for quantitative PCR in

an iCyclerMyiQ thermocycler (BioRad). Specific primers were purchased from Sigma (KiCqStart SYBR Green Primers) for the detection of the different genes. Relative expression of genes was normalized to the housekeeping gene *Actb* using the $2^{-\Delta Ct}$ method and for the expression in tendon then normalized to mean Ct value of non-treated samples using the $2^{-\Delta\Delta Ct}$ method (value shown in Figures). When expression in non-treated samples was below threshold and was non detected (ND), $2^{-\Delta Ct}$ values are shown in the Figures. To compare gene expression between wild type and knock-out mice, the fold change of $2^{-\Delta\Delta Ct}$ values of the knock-out mice was calculated with respect to the average of the $2^{-\Delta\Delta Ct}$ values of the wild type mice.

## Histopathology

Mice paws were fixed using 4% *p*-formaldehyde (Sigma-Aldrich) for at least 24 h, processed, paraffin-embedded, and sectioned in 4 µm slides. Hematoxylin and eosin stained slices were initially evaluated in a 0–3 qualitative scale, with 0 being control (healthy tendon) conditions, 1 mild, 2 medium, and 3 severe, for inflammatory infiltration, tendon cellularity and neo-vascularization grade (the median value for each of the paws was used as the final value represented in the Figures). The number of polymorphonuclear cells was quantified by counting three different fields for each sample, and these cells were identified by their nuclear morphology. The number and area of tenocytes nuclei was evaluated using the FIJI macro based on a manual threshold to select nuclei and evaluating the different parameters measured with the 'Analyze particles' tool. Sirius red staining was performed in the slides using the Picro Sirius red stain kit (Abcam) following the manufacturer's instructions and polarized light pictures (*Figure 7—figure supplement 1D*) were used to quantify the type of collagen by converting pictures to SHG color and then using either a script based on the number of pixels of each color and calculating the percentage of type I or III collagen or the CT-Fire algorithm to calculate width, length, straightness and angle of collagen fibers in these pictures (*Liu et al., 2017*). Toluidine blue staining was performed using a toluidine blue polychrome solution (Bio-Optica) for metachromatic staining of acid substances, and this staining was used for the mastocytes count. Immunohistochemistry with anti-F4/80 rat monoclonal antibody (MCA497GA, BioRad) was used to quantify macrophages by counting three different fields for each sample. All slides were examined with a Zeiss Axio Scope AX10 microscope with 20 x and 40 x objectives (Carl Zeiss) and pictures were taken with an AxioCam 506 Color (Carl Zeiss).

## Biomechanical testing of tendons

Achilles tendons were dissected following the protocol described in *Rigozzi et al., 2009*. In brief, tendons were dissected maintaining intact the calcaneus and the gastrocnemius/soleus muscles. The tendon sheaths were also maintained in order to preserve the natural anatomical structure and relative orientation of the individual tendon bundles (*Figure 7—figure supplement 3A*). Gastrocnemius/soleus muscle fibers were then cautiously removed to expose the intramuscular tendon fibers (*Figure 7—figure supplement 3B*). All mechanical tests were performed with an Autograph AG-X plus 50 N-5KN machine (Shimadzu) with a speed of 0.1 mm/s and 1 kN load head. Specimens were clamped for testing with the calcaneus mounted to approximate a neutral anatomical position (*Figure 7—figure supplement 3C*). Tendon area was calculated measuring tendon width in two segments, frontal and lateral. Tendon area and length were then used to calculate stiffness of tendons. Other parameters such as maximum force and maximum tension were also obtained. Elastic module was calculated as the slope of the curve generated representing force *vs* displacement. The slope was calculated taking into account the curve generated only between 1–2 N of force. All parameters were obtained using Trapezium X software (Shimadzu).

## Second harmonic generation microscopy

A detailed description of the second harmonic generation microscope used can be found in *Skorsetz et al., 2016*, a non-lineal optical tool (*Campagnola et al., 2001*). Unstained collagen molecules are able to generate second harmonic signal due to their natural structure with lack of a center of inversion symmetry (*Fine and Hansen, 1971*). The imaging instrument used combines a Ti:Sapphire femtosecond laser source and an inverted microscope. The laser system emits light pulses of 800 nm of wavelength at a repetition rate of 76 MHz. An XY scanning unit and a Z-motor attached to the microscope objective allow the sample to be scanned across the plane and depth location of interest. The

second harmonic generation signal from the sample propagates back through the same objective used for excitation (dry long-working distance, 20 x, 0.8 N.A.), is isolated by a short-wave pass filter (400 ± 5 nm), and finally detected by a photon counting photomultiplier module. A home-made LabView software controlled the entire system. The average power at the sample's plane was always below 100 mW. Second harmonic generation images were acquired at 2 Hz. To analyze the structural organization of the collagen fibers in the tendon, an algorithm based on the Hough transform was used (*Bueno et al., 2020*). The Hough transform is a mathematical procedure able to detect aligned segments of collagen fibers (*Figure 7—figure supplement 4*) within an image, and provide quantitative information on the degree of organization of the spatially resolved structures. On the basis of a pixel-by-pixel calculation, when a straight line is found, the corresponding polar coordinates are filed in the so-called 2D accumulator matrix. For each new detected straight line, the accumulator increases one unit. The local peaks (i.e. maximum values) in this accumulator space determine the preferential orientations found across the image. The standard deviation of these orientations is defined as the structural dispersion. A previously developed Matlab script was used for image processing (*Bueno et al., 2020*).

## Statistics

Statistical analyses were performed using GraphPad Prism 7 (Graph-Pad Software, Inc). A Shapiro-Wilk normality test was initially performed on all groups to decide the analysis type to be used. For two-group comparisons, nonparametric Mann-Whitney *U* test (without making the assumption that values are normally distributed) or the parametric unpaired *t*-test (for normal distributed data) were used to determine the statistical significance. For more than two group comparisons, one-way ANOVA test (for normal distributed data) or nonparametric Krustal-Wallis test (without making the assumption that values are normally distributed) were used to determine the statistical significance. Data are shown as mean values and error bars represent standard error from the number of independent assays indicated in the figure legend, which are also overlaid in the histograms as dot-plotting. p Value is indicated as *p < 0.05; **p < 0.01; ***p < 0.001; ****p < 0.0001; p> 0.05 not significant (*ns*).

## Acknowledgements

We thank MC Baños (IMIB-Arrixaca, Murcia, Spain) for technical assistance with molecular and cellular biology, A García Martínez (CESMAR Electromedicina) for electrode development for in vitro application of galvanic current, D Peñín Franch for helping in the development of the plugin to measure different types of collagen, FJ Ávila (University of Zaragoza, Spain) for sharing the Matlab script that was used for image processing, AJ Ortiz Ruiz (University of Murcia, Spain) for helping with tendon mechanical measurements and F Noguera and M Martínez (IMIB-Arrixaca, Murcia, Spain) for running the Hitachi ion detection system and the members of P Pelegrin's laboratory for comments and suggestions throughout this project. We also want to acknowledge the support of the SPF-animal house from IMIB-Arrixaca. Funding: AP-F was supported by MVClinic and Prim. This work was supported by grants to PP from *FEDER/Ministerio de Ciencia, Innovación y Universidades – Agencia Estatal de Investigación* (grant SAF2017-88276-R and PID2020-116709RB-I00), *Fundación Séneca* (grants 20859/PI/18 and 21081/PDC/19), and the European Research Council (ERC-2013-CoG grant 614,578 and ERC-2019-PoC grant 899636). *Ministerio de Ciencia e Innovación, Agencia Estatal de Investigación* supported JMB (grant PID2020-113919RB-I00).

## Additional information

### Competing interests

Alejandro Peñin-Franch: PhD contract was supported by MVClinic Institute and Prim. Francisco Minaya-Muñoz, Fermín Valera-Garrido: Employe of MVClinic Institute. Pablo Pelegrín: Inventor in a patent filed on March 2020 by the Fundación para la Formación e Investigación Sanitaria de la Región de Murcia (PCT/EP2020/056729) for a method to identify NLRP3-immunocompromised sepsis patients. Is consultant of Glenmark Pharmaceutical and co-founder of Viva in vitro diagnostics SL. The other authors declare that no competing interests exist.

## Funding

| Funder | Grant reference number | Author |
|---|---|---|
| Ministerio de Ciencia, Innovación y Universidades | SAF2017-88276-R | Pablo Pelegrín |
| Ministerio de Ciencia, Innovación y Universidades | PID2020-116709RB-I00 | Pablo Pelegrín |
| Fundación Séneca | 20859/PI/18 | Pablo Pelegrín |
| Fundación Séneca | 21081/PDC/19 | Pablo Pelegrín |
| European Research Council | 614578 | Pablo Pelegrín |
| European Research Council | 899636 | Pablo Pelegrín |
| Ministerio de Ciencia, Innovación y Universidades | PID2020-113919RB-I00 | Juan M Bueno |

The funders had no role in study design, data collection and interpretation, or the decision to submit the work for publication.

## Author contributions

Alejandro Peñin-Franch, Formal analysis, Investigation, Methodology, Writing – original draft; José Antonio García-Vidal, Conceptualization, Formal analysis, Investigation, Methodology; Carlos Manuel Martínez, Juan M Bueno, Formal analysis, Investigation, Methodology, Writing – review and editing; Pilar Escolar-Reina, Ana I Gómez, Investigation; Rosa M Martínez-Ojeda, Investigation, Methodology; Francisco Minaya-Muñoz, Fermín Valera-Garrido, Conceptualization, Funding acquisition, Methodology, Resources, Writing – review and editing; Francesc Medina-Mirapeix, Conceptualization, Funding acquisition, Methodology, Resources, Supervision, Writing – review and editing; Pablo Pelegrín, Conceptualization, Formal analysis, Funding acquisition, Supervision, Writing – original draft, Writing – review and editing

## Author ORCIDs

Alejandro Peñin-Franch (iD) http://orcid.org/0000-0001-5329-1671
Carlos Manuel Martínez (iD) http://orcid.org/0000-0003-3307-1326
Fermín Valera-Garrido (iD) http://orcid.org/0000-0002-8867-2519
Pablo Pelegrín (iD) http://orcid.org/0000-0002-9688-1804

## Ethics

We present non-invasive ultrasound scanning images from routinely clinical following from a single patient. The data is completely anonymised and from the normal clinical routine practice.
All experimental protocols for animal handling were refined and approved by the local animal research ethical committee (references 241/2016 and 541/2019) and Animal Health Service of the General Directorate of Fishing and Farming of the Council of Murcia (Servicio de Sanidad Animal, Dirección General de Ganadería y Pesca, Consejería de Agricultura y Agua de la Región de Murcia, reference A13160702). Animal experimentation was performed in strict accordance with the Hospital Clínico Universitario Vírgen de la Arrixaca animal experimentation guidelines, and the Spanish national (RD 1201/2005 and Law 32/2007) and EU (86/609/EEC and 2010/63/EU) legislation.

## Decision letter and Author response

Decision letter https://doi.org/10.7554/eLife.73675.sa1
Author response https://doi.org/10.7554/eLife.73675.sa2

# Additional files

## Supplementary files
• Transparent reporting form

## Data availability

All data generated or analysed during this study are included in the manuscript and supporting file; Source Data files have been provided for Figures 1 to 7.

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
