## [Editor Report]

This study investigated the mechanisms that underlie chronic tendinopathies. The authors identified the role of galvanic current on cellular death and inflammation in vitro and its association with tendinopathy in vivo. They showed that galvanic current induces NLRP3 inflammasome-driven low-grade inflammatory response, which promotes collagen-mediated regeneration of the tendon. This work may lead to a therapeutic strategy to treat chronic tendinopathies.

---

## [Decision Letter]

**Decision letter after peer review:**

Thank you for submitting your article "Galvanic current activates the NLRP3 inflammasome to promote type I collagen production in tendon" for consideration by *eLife*. Your article has been reviewed by 2 peer reviewers, and the evaluation has been overseen by a Reviewing Editor and Mone Zaidi as the Senior Editor. The reviewers have opted to remain anonymous.

Essential revisions:

Recommendations for the authors are listed below by both reviewers and should be considered for revision.

*Reviewer #2:*

The in vitro studies were major strengths of this manuscript, such as the rigorous evaluation of cell membrane damage (using LDH release) and cell death (using Yo-Pro-1). These findings could be better described in text and in the figure, as "LDH release" could be better communicated as "cell membrane damage" and Yo-Pro-1 could be described as "cell viability" for clarity to a more general audience.

Using in vitro knockout experiments, this study showed that galvanic current activates the NLRP3 inflammasome and this inflammatory response could be abbrogated by increased extracellular KCl. in vivo, galvanic current application also led to an increased expression of inflammatory markers and accumulation of inflammatory cells at acute post-puncture time points, which was enhanced by knockdown of NLRP3. However, the inflammasome response transcriptionally was diminished (using qRTPCR) by knockdown of NLRP3. These findings are a bit confusing and challenging to interpret.

The inclusion of functional and structural assessments (e.g., biomechanical testing, collagen alignment, collagen damage) associated with tendon strength and recovery would be valuable post-treatment to better understand how percutaneous electrolysis influences the tendon structure. This would strengthen the in vivo findings, which are primarily based on zoomed-in histological outcomes and biochemical assays of collagen.

Figure 7 includes patient data from a single patient, over time, as well as quantitative PCR data from NLRP3 ko mice. This figure is a bit confusing and difficult to understand on its own, considering it includes data from both clinical and animal studies without clear annotations in the figure to describe what is what. Additionally, the clinical data from a single patient is somewhat limited, and generally these data would be better supported and more rigorous if additional patients were included.

Lastly, the patient data are from the elbow, whereas the mouse data are from the calcaneal tendon, and there may be anatomical and biomechanical differences in how these tendons response to galvanic current. Explanations in the discussion could help here.

In general, this paper could use a more clearly articulated introduction describing (1) what is galvanic current therapy, and when and why is it used? (2) is there a known or unknown role of inflammation in improving or abbrogating tendon dysfunction, pain, and health? and (3) what about the NLRP3 inflammasome is unique or different from other mechanisms (and why would galvanic current treatment uniquely target this particular inflammasome)? Structuring the introduction to sequentially tell the story in this way will help to establish the premise of this work better.

Some of the data included in supplemental could be in the main text figures, and vice versa, considering each figure has a specific "take home" message that is being delivered that adds to the story. Some of the data and labeling in the figures (e.g. Figure 3) need to be revised as not all panels are clearly communicating their content (Figure 3F doesnt have a legend; and the bar chart beneath Figure 3F doesn’t have x-axis labels). Zoomed-in histology in Figure 4 may be misleading and a larger field of view to visualize tendon structure would be informative. Dot plots for Figure 4B/D would be helpful and in line with how all other figures in this manuscript are plotted.

*Reviewer #3:*

Peñín-Franch et al. report that galvanic current induces NLRP3 inflammasome-driven low-grade inflammatory response, which promotes collagen-mediated regeneration of the tendon. The data are clearly presented, and the narrative is in general comprehensible. However, the story is convoluted as in vitro data do not fully match in vivo findings, the mechanism of action of galvanic current through K^+^ efflux is not convincing, the rationale on the focus of the experiments on bone marrow-derived macrophages instead tendon-resident macrophages is not provided, the omission of neutrophils for mechanistic insight, and the other inflammasomes are not explored. These concerns are outlined below.

Figure 1A: Concerns that cells in these experimental conditions are not very responsible to LPS as the induction of Cox2 expression by a high dose of the endotoxin (1 µg/ml) is modest.

Figure 1B: Need to show the expression of genes whose expression is not reduced by galvanic application to rule nonspecific effects.

Figure 1C: Why is IL-18 secretion not analyzed?

Figure 2A: Cells lacking the other inflammasomes such as AIM2, NLRP1, and NLRC4, some of which may not require ASC need to be studied.

The authors found that galvanic current failed to decrease intracellular K^+^ levels; yet, they still argued that galvanic current affect K^+^ efflux to some extent. This is confusing.

Figure 2A-D: LPS needs to be added to all graphs. The legends in B-D does not specify if cells were treated with LPS or not.

Figure 3A and B: Since galvanic current does not induced pyroptosis through NLRP3, the authors need absolutely to show the effect of the current on production of IL-18, another effector of the pathway.

The statement “Two impacts of galvanic currents of different intensities (3, 6, 12 mA) for a period of 6 seconds (conditions that induce IL-1b release) were only inducing a significant, but slightly increase of cell death (Figure 3A) (lines 157-162) is awkward.

The statement "This increase in cell death was not associated with the activation of the inflammasome, since it was also present in macrophages deficient on NLRP3, ASC or caspase-1/11 (Figure 3B), suggesting that was independently of pyroptosis” is another reason to check the other inflammasomes.

Figure 4A showing that galvanic current increases the number of PMNs provides a strong argument for studying the effects of this treatment on cytokine and LDH release by these cells.

Figure 5: What is the relevance of these results since galvanic current increases the expression of some of these genes in vitro macrophages without affecting their protein levels (Figure 1)?

A great deal of the discussion revolves around the background of the inflammasomes (lines 249- 263) instead of discussing the results.

The discussion around the unexpected increased in PMN number in Nlrp3-deficient mice is not well articulated.

---

## [Author Response]

Reviewer #2:The in vitro studies were major strengths of this manuscript, such as the rigorous evaluation of cell membrane damage (using LDH release) and cell death (using Yo-Pro-1). These findings could be better described in text and in the figure, as “LDH release” could be better communicated as “cell membrane damage” and Yo-Pro-1 could be described as “cell viability” for clarity to a more general audience.

We have changed the description of LDH release and Yo-Pro-1 uptake as suggested by the reviewer. Please see Figure 3 legend and page 8, lines 181-183:

“…we next assessed pyroptosis by means of Yo-Pro-1 uptake to cells, to measure plasma membrane pore formation and cell viability, and LDH leakage from the cell, to determine plasma membrane damage”.

Using in vitro knockout experiments, this study showed that galvanic current activates the NLRP3 inflammasome and this inflammatory response could be brogated by increased extracellular KCl. in vivo, galvanic current application also led to an increased expression of inflammatory markers and accumulation of inflammatory cells at acute post-puncture time points, which was enhanced by knockdown of NLRP3. However, the inflammasome response transcriptionally was diminished (using qRTPCR) by knockdown of NLRP3. These findings are a bit confusing and challenging to interpret.

In vivo determination of NLRP3 inflammasome activation is a challenge issue and mainly relying on the use of NLRP3-KO mice. Our results show that in vitro there are strong evidences to conclude that galvanic current activates the NLRP3 inflammasome, but in vivo seems that NLRP3 deficiency affects some inflammatory parameters (expression of Il1b, il1rn and Cxcl10), as opposed of the potentiation of other responses (infiltration of polymorphonuclear cells). At structural and functional level, the lack of NLRP3 has an effect on the tendons treated with galvanic currents, so we assume that the inflammatory programme controlled by NLRP3 has a role in the effects of galvanic currents applicated in vivo.

The fact of a response to galvanic currents dependent on NLRP3 and a response independent to NLRP3 in vivo was depicted in the diagram of Figure 7—figure supplement 2.

Our data support the notion that in the tendons, NLRP3 do not affect PMN infiltration, but overall control IL-1-drived inflammatory response. We have rephrased discussion to clarify and ease with data interpretation, see page 14, lines 331-334:

“NLRP3 was important to induce an inflammatory response in vivo with elevation of different cytokines including Il1b or Cxcl10, that conditioned the structure and functions of the treated tendons. […] This highlight that NLRP3 induced by galvanic current is able to control a specific inflammatory programme in vivo, but it probably do not affect the IL-6-infiltration of polymorphonuclear cells in tendons”.

The inclusion of functional and structural assessments (e.g., biomechanical testing, collagen alignment, collagen damage) associated with tendon strength and recovery would be valuable post-treatment to better understand how percutaneous electrolysis influences the tendon structure. This would strengthen the in vivo findings, which are primarily based on zoomed-in histological outcomes and biochemical assays of collagen.

As suggested by the reviewer, we have now performed new experiments of tendon function and structure. By one hand we measured biomechanical properties of the tendons by using an universal testing machine for tensile tests, and by other we measured the structural organization of the collagen fibers in the tendons by employing state-of-the-art second harmonic generation microscopy.

We found that galvanic current increase stiffness of the tendon via NLRP3-inflammasome, but did not significantly affected maximum force of the tendon (new data on Figure 7E). The increase of the stiffness of the tendon is related with an increase in the resistance (i.e. PMID: 34483334 and 34510279), which is reduced for example during ageing and resulting in weaker tendons (i.e. PMID: 34274923).

This was further supported by studying the alignment of collagen fibers by second harmonic generation microscopy and applicating an algorithm based on the Hough transform, where structural dispersion of collagen fibers was higher in Nlrp3^-/-^ tendons after galvanic current application compared to tendons from wild type mice (new data on Figure 7D and Figure 7— figure supplement 1E).

We have included these data in results, discussion and methods, showing that galvanic current application lead to a regeneration of the tendon via NLRP3 implication by increasing type I collagen, the arrangement of the collagen fibers and increasing the resistance of the tendon to change in length. These results clearly strength the conclusions of the study over our previous histological data.

Figure 7 includes patient data from a single patient, over time, as well as quantitative PCR data from NLRP3 ko mice. This figure is a bit confusing and difficult to understand on its own, considering it includes data from both clinical and animal studies without clear annotations in the figure to describe what is what. Additionally, the clinical data from a single patient is somewhat limited, and generally these data would be better supported and more rigorous if additional patients were included.

We agree with the reviewer and the inclusion of a single patient was an example of the actual application and outcomes of the technique in clinics. Studies including additional patients have been already published (i.e. PMID: 25122629, 27854082, 32545583) and referenced in the text. Taking reviewer advice we moved human data to Figure 7—figure supplement 1A and clearly expressing in the text that is the therapeutic outcome of the validated technique in clinics, referencing other studies where higher cohorts of patients are presented. Now, figure 7 only show mouse-related data with the increase panels showing functional and structural assessments of the tendons as explained in the previous point.

Lastly, the patient data are from the elbow, whereas the mouse data are from the calcaneal tendon, and there may be anatomical and biomechanical differences in how these tendons response to galvanic current. Explanations in the discussion could help here.

As suggested we have discussed this issue in the revised manuscript. We cannot exclude differences among calcaneal and epicondyle tendons responding to galvanic current application, the calcaneal tendon in the mice was the only one big enough to apply the technique with a ultrasound-guided needling, and therefore we had a procedure limitation. However, clinical application of galvanic current has been successfully applicated in different body tendons, including supraspinatus tendinopathy (PMID: 32545583), patellar tendinopathy (PMID: 27854082) and lateral epicondylitis (PMID: 25122629). In all cases, a similar clinical benefit to galvanic current was found. This is now discussed in the revised manuscript in page 15, lines 353-356:

“A limitation of our study is that the animal model used restrict us to employ the ultrasound guided puncture to applicate galvanic current to the calcaneal tendon of mice, as is the larger accessible tendon, whereas therapeutically application of galvanic current in humans have been applicated in supraspinatus (Rodríguez-Huguet et al., 2020), patellar (Abat et al., 2016) and lateral epicondyle (Valera-Garrido et al., 2014) tendons. […] In fact, there have been described species differences in the inflammatory response of tenocytes (Oreff et al., 2021), although mice was presented a high overall similarities of tenocyte response when compared to human tenocytes, additional animal models applicating galvanic current in the presence of specific NLRP3 blockers, as MCC950, would be required”.

In general, this paper could use a more clearly articulated introduction describing (1) what is galvanic current therapy, and when and why is it used? (2) is there a known or unknown role of inflammation in improving or abrogating tendon dysfunction, pain, and health? and (3) what about the NLRP3 inflammasome is unique or different from other mechanisms (and why would galvanic current treatment uniquely target this particular inflammasome)? Structuring the introduction to sequentially tell the story in this way will help to establish the premise of this work better.

We have structured the introduction as suggested by the reviewer, which we believe has improved the clarity of the manuscript.

Some of the data included in supplemental could be in the main text figures, and vice versa, considering each figure has a specific "take home" message that is being delivered that adds to the story. Some of the data and labeling in the figures (e.g. Figure 3) need to be revised as not all panels are clearly communicating their content (Figure 3F doesnt have a legend; and the bar chart beneath Figure 3F doesnt have x-axis labels). Zoomed-in histology in Figure 4 may be misleading and a larger field of view to visualize tendon structure would be informative. Dot plots for FIgure 4B/D would be helpful and in line with how all other figures in this manuscript are plotted.

As suggested, we have fixed Figure 3F panel (the legend was on the right size of the graph) and we have added a larger view of the tendon in new Figure 4—figure supplement 1C. According to the reviewer, we also have moved Figure 7A to supplemental materials (now as Figure 7—figure supplement 1A) and part of the Figure 2—figure supplement 1B to main figures (now Figure 2E), as now we have included more galvanic current conditions. If the reviewer thinks additional panels could be swapped between main and supplementary figures, we will be happy to evaluate these further changes. Regarding dot-plot representation for Figure 4B and D, since it is a kinetic of different treatments that overlap in the time points assessed, if we do dot-plot representation, the graph is not clear as the dots of the different treatments overlap each other (Author response image 1) .

**Author response image 1. sa2fig1:** 

Therefore, we have opted to show this dot raw data plot representation in Figure 4 – source data file 2, but separating the different treatments.

Reviewer #3:Peñín-Franch et al. report that galvanic current induces NLRP3 inflammasome-driven low-grade inflammatory response, which promotes collagen-mediated regeneration of the tendon. The data are clearly presented, and the narrative is in general comprehensible. However, the story is convoluted as in vitro data do not fully match in vivo findings, the mechanism of action of galvanic current through K^+^ efflux is not convincing, the rationale on the focus of the experiments on bone marrow-derived macrophages instead tendon-resident macrophages is not provided, the omission of neutrophils for mechanistic insight, and the other inflammasomes are not explored. These concerns are outlined below.Figure 1A: Concerns that cells in these experimental conditions are not very responsible to LPS as the induction of Cox2 expression by a high dose of the endotoxin (1 µg/ml) is modest.

We would like to highlight that stimulation time with LPS in our study was 2h, and usually longer times (up to 24h) are normally used and that could explain why Cox2 expression is “modest”. In any case, the data shown in Figure 1A is presented as relative expression values (2^-DCt^) to be able to compare relative expression to galvanic current application. If we calculate fold increase (2^-DDCt^) of Cox2 gene we found an average of 107 with values from individual experiments ranging from 30 to 322 (Author response image 2).

Other similar studies reporting Cox2 expression levels on mouse macrophages report higher Cox2 induction (400 fold increase) but as stated used longer times of LPS (2h our study vs 24h) (PMID 26639663). However, other studies using 4h LPS stimulation but with a lower LPS dose (1µg/ml our study vs 100 ng/ml) reported a similar fold increase with an average of ∼150 (PMID: 25116357, 28962083). Therefore, this show that the macrophage used in our study were robustly responding to LPS similarly than in other studies. In any case, this is not the central point of the study and will not affect the overall conclusion.

Figure 1B: Need to show the expression of genes whose expression is not reduced by galvanic application to rule nonspecific effects.

As suggested by the reviewer, we do now present the expression of the M2 marker Ym1 in Figure 1B that is not reduced by galvanic current application. However, please also note that in Figure 1A we show that the expression of Tnfa and Il1b do not change by galvanic current application.

Figure 1C: Why is IL-18 secretion not analyzed?

In Figure 1C we aimed to validate gene expression of cytokines measured by qPCR in panel 1A, and this was the reason why we did not show IL-18. We now show IL-18 release as a inflammasome-related cytokine in the new Figure 2C and found that as expected IL-18 was also being released by galvanic current in a NLRP3 dependent manner, since NLRP3 KO macrophages (new Figure 2C) and the use of MCC950 (new Figure 3C) also blocked IL-18 release.

Figure 2A: Cells lacking the other inflammasomes such as AIM2, NLRP1, and NLRC4, some of which may not require ASC need to be studied.

We appreciate reviewer comment, however our data using both NLRP3 KO macrophages and the specific NLRP3 blocker MCC950 clearly show that NLRP3 is the inflammasome activated by galvanic current. If other inflammasome sensor would be also activated after galvanic current application, NLRP3-KO macrophages or MCC950-treated macrophages would still releasing IL-1b and IL-18. Therefore, we strongly believe that the study of macrophages lacking other inflammasome sensors would not give further insights into this study and are not needed to be studied.

In any case, and taking reviewer suggestion, we now have increased the insights that NLRP3 is the activated inflammasome after galvanic current application by using a recombinant system where NLRP3-YFP is expressed in HEK293. This system has been widely used to assess NLRP3 activation by us (i.e. PMID 22981536, 31086329, 34524838) and other groups (i.e. PMID 30487600). Galvanic current was able to increase the percentage of cells with an intracellular NLRP3 forming a punctum, and this was reverted by the use of the specific blocker MCC950. See data in new Figure 2G.

The authors found that galvanic current failed to decrease intracellular K^+^ levels; yet, they still argued that galvanic current affect K^+^ efflux to some extent. This is confusing.

We have now measured intracellular K^+^ in macrophages receiving increasing doses of galvanic current (where a robust IL-1b release was found), and now we were able to find K^+^ efflux from the macrophages (see new Figure 2E), correlating with the ability of high extracellular K^+^ to block IL-1b release after galvanic current application.

Figure 2A-D: LPS needs to be added to all graphs. The legends in B-D does not specify if cells were treated with LPS or not.

As suggested, we added LPS treatment to the graph legends in figure 2. This was also extended to figure 3.

Figure 3A and B: Since galvanic current does not induced pyroptosis through NLRP3, the authors need absolutely to show the effect of the current on production of IL-18, another effector of the pathway.

We now show IL-18 release, that similarly to IL-1b, was induced by galvanic current and blocked by MCC950 (see new Figure 3C). The fact that galvanic current induces cell death independently of NLRP3 and also caspase-1 and caspase-11 (previous Figure 3B,C, now Figure 3B,D) suggest that this cell death is not pyroptosis and probably is a direct consequence of applicating the current to the cell.

The statement "Two impacts of galvanic currents of different intensities (3, 6, 12 mA) for a period of 6 seconds (conditions that induce IL-1b release) were only inducing a significant, but slightly increase of cell death (Figure 3A) (lines 157-162) is awkward.

We apologize, as we did not explain correctly. We have revised the phrase (page 8, lines 183-186):

“Two impacts of galvanic currents of different intensities (3, 6, 12 mA) for a period of 6 seconds (conditions that induce IL-1b release as we show in Figure 1C) were only inducing a slightly increase of cell death (Figure 3A).”

The statement "This increase in cell death was not associated with the activation of the inflammasome, since it was also present in macrophages deficient on NLRP3, ASC or caspase-1/11 (Figure 3B), suggesting that was independently of pyroptosis" is another reason to check the other inflammasomes.

Other inflammasomes will also signal through caspase-1 or caspase-11 (if there is an activation of the non-canonical inflammasome). The fact that macrophages doble knockout in caspase-1 and caspase-11 do not impair cell death induced by galvanic current strongly suggest that no other inflammasome is activated. For example, if NLRC4 would be activated, the casp1/11 KO macrophages would give a phenotype where LDH release was blocked. Therefore, we have strong results concluding that the application of galvanic current could kill the cell independently of caspase-1/11, and therefore independent of any inflammasome (Figure 3B).

Figure 4A showing that galvanic current increases the number of PMNs provides a strong argument for studying the effects of this treatment on cytokine and LDH release by these cells.

We found reviewer comment very pertinent and following his/her advise we have now assessed IL-1b release and LDH on bone marrow isolated neutrophils upon galvanic current application. However, galvanic current application was not inducing the release of IL-1b (see new Figure 4—figure supplement 1B).

Figure 5: What is the relevance of these results since galvanic current increases the expression of some of these genes in vitro macrophages without affecting their protein levels (Figure 1)?

We aimed to characterize the inflammatory response in the tendon, and although we completely agree with reviewer comment for IL-6 measured in figure 1 and 5, we were unable to detect cytokines by ELISA determination in the tendon as the amount of tissue is very small. Gene expression by qPCR was the technique together histopathology to characterize the inflammatory response. We have explained in the results the limitations of this technique, and contrasted with the fact shown in Figure 1 than despite increase of mRNA level, increase of cytokine secretion could not correlate. See page 14, lines 331-334:

“NLRP3 was important to induce an inflammatory response in vivo with elevation of different cytokines including Il1b or Cxcl10, that conditioned the structure and functions of the treated tendons. […] This highlight that NLRP3 induced by galvanic current is able to control a specific inflammatory programme in vivo, but it probably do not affect the IL-6-infiltration of polymorphonuclear cells in tendons”.

A great deal of the discussion revolves around the background of the inflammasomes (lines 249-263) instead of discussing the results.

We have fixed the first paragraph of the discussion as suggested by the reviewer to remove the background of the inflammasome, see page 13.

The discussion around the unexpected increased in PMN number in Nlrp3-deficient mice is not well articulated.

As suggested by the reviewer we have modified this part of the revision. See page 14 lines 333-336:

“However, NLRP3 deficiency do not affect Il6 production and the infiltration of polymorphonuclear cells when galvanic currents were applicated in vivo. This highlight that NLRP3 induced by galvanic current is able to control a specific inflammatory programme in vivo, but it probably do not affect the IL-6-infiltration of polymorphonuclear cells in tendons”.